# Change in the potential snowfall phenology: past, present, and future in the Chinese Tianshan mountainous region, Central Asia

**Xuemei Li[1,2,3], Xinyu Liu[1,2,3], Kaixin Zhao[1,2,3], Xu Zhang[1,2,3], and Lanhai Li[4,5]**

[1]Faculty of Geomatics, Lanzhou Jiaotong University, Lanzhou 730070, China;
[2]National-Local Joint Engineering Research Center of Technologies and Applications for National Geographic State Monitoring, Lanzhou 730070, China;
[3]Gansu Provincial Engineering Laboratory for National Geographic State Monitoring, Lanzhou 730070, China;
[4]State Key Laboratory of Desert and Oasis Ecology, Xinjiang Institute of Ecology and Geography, Chinese Academy of Sciences, Urumqi 830011, China;

[5]Research Center for Ecology and Environment of Central Asia, Chinese Academy of Sciences, Urumqi 830011, China

Correspondence: Xuemei Li (lixuemei@mail.lzjtu.cn)

**Abstract.** The acceleration of climate warming has led to a faster solid-liquid water cycle and a decrease in solid water storage in cold regions of the Earth. Although snowfall is the most critical input for the cryosphere, the phenology of snowfall, or potential snowfall phenology (PSP), has not been thoroughly studied, and there is a lack of indicators for PSP. For this reason, we have proposed three innovative indicators, namely, the start of potential snowfall season (SPSS), the end of potential snowfall season (EPSS), and the length of potential snowfall season (LPSS), to characterize the PSP. We then explored the spatial-temporal variation of all three PSP indicators past, present, and future across the Chinese Tianshan mountainous region (CTMR) based on the observed daily air temperature from 26 meteorological stations during 1961-2017/2020 combined with 14 models data from CMIP6 (the Phase 6 of the Coupled Model Intercomparison Project) under four different scenarios (SSP126, SSP245, SSP370, and SSP585) during 2021-2100. The study showed that the SPSS, EPSS, and LPSS indicators could accurately describe the PSP characteristics across the study area. In the past and present, the potential snowfall season started on November 2nd, ended on March 18th, and lasted for about four and a half months across the CTMR on average. During 1961-2017/2020, the rate of advancing EPSS (-1.6 days/10a) was faster than that of postponing SPSS (1.2 days/10a). It was also found that there was a significant delay in the starting time (2-13 days) and advancement in the ending time (1-13 days), respectively, resulting in a reduction of 3-26 days for the LPSS. The potential snowfall season started earlier, ended later, and lasted longer in the north and center compared to the south. Similarly, the SPSS, EPSS, and LPSS indicators is also expected to vary under the four emission scenarios during 2021-2100. Under the highest emission scenario, SSP585, the starting time is expected to be postponed by up to 41 days, while the ending time is expected to be advanced by up to 23 days across the study area. This change is expected to reduce the length of the potential snowfall season by up to 61 days (about 2 months), and the length of the potential snowfall season will only last 2 and a half months in the 2100s under the SSP585 scenario. The length of the potential snowfall season in the west and southwest of the CTMR will be compressed by more days due to more delayed starting time and advanced ending time under all four scenarios. This suggests that with constant snowfall intensity, annual total snowfall may decrease, including amount and frequency, leading to a reduction in snow cover or mass, which will ultimately contribute to more rapid warming through the lower reflectivity to solar radiation. This research

45    provides new insights into capturing the potential snowfall phenology in the alpine region and can be easily extended to other snow-dominated areas worldwide. It can also help inform snowfall monitoring and early warning for solid water resources.

**Keywords:** Acceleration of climate warming, potential snowfall phenology, spatial-temporal variation, Chinese Tianshan mountainous region

## 1 Introduction

Snowfall, as a solid form of precipitation, exerts a significant impact on human society, ecological environment, and hydrological processes in mountainous areas, making it a crucial water resource (Barnett et al., 2005; Jonas et al., 2008; Krasting, 2008; McAfee et al., 2014; Bai et al., 2019; Zhang et al., 2019; Tamang et al., 2020). The warming rate in the mountains is about twice that of the rest of the planet (Sabine et al., 2022). Due to the significant variation of topography, massive solid water storage (glacier and snowpack), and its high sensitivity to climate change in the mountains, mountainous regions are particularly vulnerable to increasing temperature and changing precipitation patterns (Dedieu et al., 2014; Piazza et al., 2014; Roux et al., 2021). Hence, mountains are often referred to as water towers of the world and outposts of global climate change (Sorg et al., 2012; Huss et al., 2017; Immerzeel et al., 2020).

One of the most prominent impacts of climate warming has been a shift from snow to rain, including frequent occurrence of rain on snow events in temperate and cold regions across the globe (Knowles et. al., 2006; Trenberth, 2011, Jennings and Molotch, 2019). While it is estimated that annually 1773 $km^3$ of snow (about 5% of the global snowfall accumulations) could fall over international mountains on average, rapid climate warming is reducing snowfall frequencies and fractions over many regions crucial to water resources and the global climate system (Mankin et al., 2015; Harpold et al., 2017; Bintanja and Andry, 2017). This is leading to predicted reductions in snow mass of up to 25% over the next 10 to 30 years (Daloz et al., 2020; Hock et al., 2022). Furthermore, changes in snowfall phenology, such as the start of the snowfall season, the end of the snowfall season, and the length of the snowfall season, have been observed. Less snowfall in winter was caused by higher surface temperature due to increased greenhouse gas emissions (Sun et al., 2016). In late winter, the increased occurrence of above-zero temperatures reduces the fraction of snowfall and enhances snow-melting in the mid-winter (Raisanen, 2016). Postponed snowfall occurrence and advanced snowfall ending have also been observed across the Eurasian continent (Bai et al., 2019; Lin and Chen, 2022).

The annual variation of air temperature in the Northern Hemisphere adheres to a specific pattern primarily influenced by solar radiation, wherein air temperature gradually increases from the beginning of the year until the middle, and subsequently decreases steadily until the end of the year (Marshall and Plumb, 2008). Precipitation falls on the ground in various phases, such as rainfall, snowfall, and sleet, each having a significant impact on the surface runoff and energy balance (Loth et al., 1993; Han et al., 2018). When the air temperature reaches rain-snow threshold (RST), precipitation falls as rain and snow with equal frequency, while above the threshold, precipitation falls primarily as rain and below primarily as snow (Jennings et al., 2018). Consequently, with air temperature below the RST in winter, there is a possibility of potential snowfall if all other conditions (water vapor, air pressure, condensation nodules etc.) are satisfied. The potential snowfall season could cover the period when the air temperature is below the RST within two consecutive years, reflecting the intra-annual fluctuation of air temperature, timing

allocation and capacity of snowfall, as well as water and energy balance in a region. A shorter potential snowfall season would result in an expanded potential rainfall season, thereby changing the potential water and energy needed by snowfall. Potential snowfall phenology (PSP), the start of potential snowfall season (SPSS), the end of potential snowfall season (EPSS), and the length of potential snowfall season (LPSS) are utilized to identify the possible onset, end, and duration of snowfall. The advancement or delay of SPSS indicates that potential snowfall arrives earlier or later in late-autumn or early-winter, affecting accumulation and storing of solid water resource as snow cover. Similarly, the advancement or delay of EPSS implies that potential snowfall ends earlier or later in late-winter or early-spring, which is likely to alter the snow-melting, snow albedo, and groundwater and streamflow dynamics due to diminished and more ephemeral snowpacks (Siirila-Woodburn et al., 2021) in mountainous areas such as the Chinese Tianshan mountainous region (CTMR).

The CTMR represents a prototypical alpine area with considerable topographic heterogeneity (Li et al., 2022). Regional warming amplification and altitude warming amplification in the CTMR were identified (Gao et al., 2021). Mean annual snowfall is expected to decrease by 26.5% during the period of 2070-2099 under the RCP8.5 scenario in the CTMR (Yang et al., 2017). Snowfall is a critical resource in the alpine region of Asia, and its sensitivity to climate change has been well-established (Kapnick et al., 2014). Air temperature is a crucial driving factor in the snowfall variation and the transition from snowfall to rainfall in the CTMR (Zhang et al., 2019; Ren et al., 2020; Ren et al., 2022). In Northwestern China, including the CTMR, a semi-sinusoid curve is capable of accurately fitting the intra-annual distribution of air temperature (Li et al., 2015). Similar to the diverse RSTs observed across the Northern Hemisphere (Jennings et al., 2018), precipitation phase records during 1950s-1979 can be utilized to obtain spatially inconsistent RSTs in the CTMR through the approach of Dai (Dai, 2008; Zhang et al, 2017).

Quantifying changes in the PSP is crucial for enhancing our comprehension of current and future climate change, which in turn facilitates the development of adaptation policies. However, few technical studies have focused on the PSP and its variability in the CTMR. Therefore, to address these knowledge gaps, we first define three indicators of the PSP in combination with the semi-sinusoid curve and RST line. Subsequently, we explore the variation of the PSP past, present, and future using available meteorological station observations in the CTMR and data from 14 models in CMIP6 (Phase 6 of the Coupled Model Intercomparison Project) under four distinct scenarios. The following section of the paper provides further details regarding the study area, data, and methods, including the definition of three indicators of the PSP. Then, we evaluate changes in the PSP throughout the CTMR in Section 3. The paper ends with the discussion and conclusions.

## 2 Study area, data, and methods

### 2.1 Study area

Situated in the heart of the Eurasian continent, Tianshan mountain is the largest mountain system in Central Asia, spanning about 2500 km in length and approximately 250-350 km in width, covering an area of $8\times10^5$ km$^2$ and extending from Uzbekistan to Kyrgyzstan, southeastern Kazakhstan, and Xinjiang (China) (Aizen et al., 1997; Yang et al., 2019; Li et al., 2020). The CTMR, located within Xinjiang, China, comprises about $5.7\times10^5$ km$^2$ (about 1700 km in length) and accounts for 34.5% of the total area of Xinjiang Uygur Autonomous Region (Hu, 2004; Li et

al., 2020) (see Figure 1). The average altitude of the mountain ridge is 4000 m.a.s.l., with the highest peak being Tomor (7435 m a.s.l.). The CTMR exhibits a typical alpine mountainous environment with a continental climate that features significant seasonal differences. Mainly influenced by the westerly circulation and topography, the CTMR receives abundant precipitation and is considered to the main source of water resources in Xinjiang. The annual mean air temperature and the annual precipitation are 7.7 ℃ and 189.58 mm, respectively (Li et al., 2016; Li et al., 2020). The warming trend in the CTMR is widespread and characterized by a significant seasonal variation (Li et al., 2022). Snowfall in the CTMR is primarily observed from November to February, while rainfall, snowfall, or sleet coexist in March and April. In recent years, both snowfall and rainfall in the CTMR have displayed a significant increasing trend, with the growth rate of rainfall being greater than that of snowfall, and S/P (the fraction of snowfall to precipitation) showing a significant decreasing trend (Guo and Li, 2015; Zhang et al., 2019). Winter snowfall can reach 84.53 mm with a slight increase (Yang et al., 2022). Moreover, there has been a noticeable shift from solid to liquid precipitation, with the frequency of rainfall increasing and that of snowfall decreasing (Tian et al., 2020; Li, 2021).

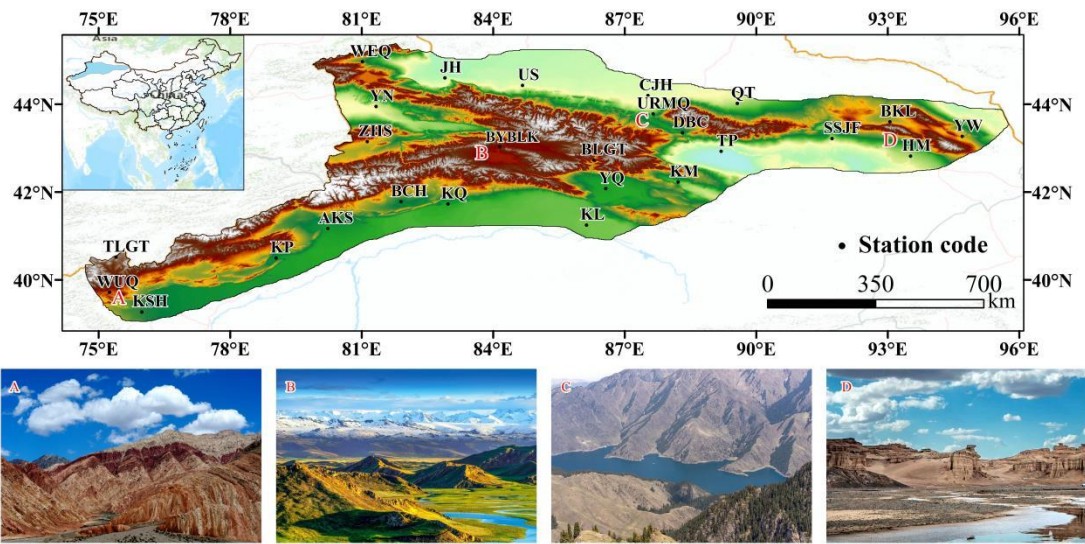

**Figure 1.** Distribution of meteorological stations in the CTMR. The full name and descriptive information of all meteorological stations selected can refer to Table 1. (A-D) shows photos taken on the spot from WeChat Official Account, Geographical Commune and Internet (https://cn.bing.com/)

## 2.2 Data
### 2.2.1 Observed historical daily air temperature data
This study used the observed daily air temperature data covering 1961-2017/2020 from 26 meteorological stations in the CTMR (see Figure 1), provided by China Meteorological Data Service Centre (http://data.cma.cn). Prior to release, the data quality was meticulously checked, and homogeneity tests were conducted, as described in previous studies (Jiang et al., 2009; Li et al., 2015; Li et al., 2020). Subsequently, the daily air temperature data were subjected to a quality control process involving visualizing individual stations or grid records to identify outliers, which were either removed or corrected. Only a small fraction of the data required correction, and missing daily values, which accounted for less than 2% of the total data, were estimated for a given day by extrapolating the average value of the data from the one or two preceding and following records. Time series with more than one year of missing data from meteorological

stations were excluded from this study. Finally, daily air temperature data from 12 meteorological stations during 1961-2017 and from 14 meteorological stations during 1961-2020 were used for

analysis. Furthermore, precipitation phase after 1979 in the CTMR, and even across China, is not recorded (Ding et al., 2014). Visual observer reports of daily precipitation phase are available only during 1950s-1979 from 26 meteorological stations across the CTMR, with a total number of records of 237115. Although values of RST or proportions were not included in the observational data, we used daily precipitation phase data to calculate RSTs using two methods: the frequency

intersection method and the probability guarantee method (Zhang et al., 2017), which evolved from the approach of Dai (2008) and Jennings et al. (2018). The average value of RSTs computed by these two methods for each station was used as the threshold to differentiate potential rainfall and snowfall. The average accuracy rates for snowfall and rainfall during 1950s-1979 were 0.96 and 0.94, respectively. The details of methods could refer to work from Dai (2008) and Zhang et al.

(2017). Figure 2 and Table 1 presented accuracy rate of discriminating snowfall and rainfall using RST value for each meteorological station in the CTMR and the details of the 26 meteorological stations, respectively.

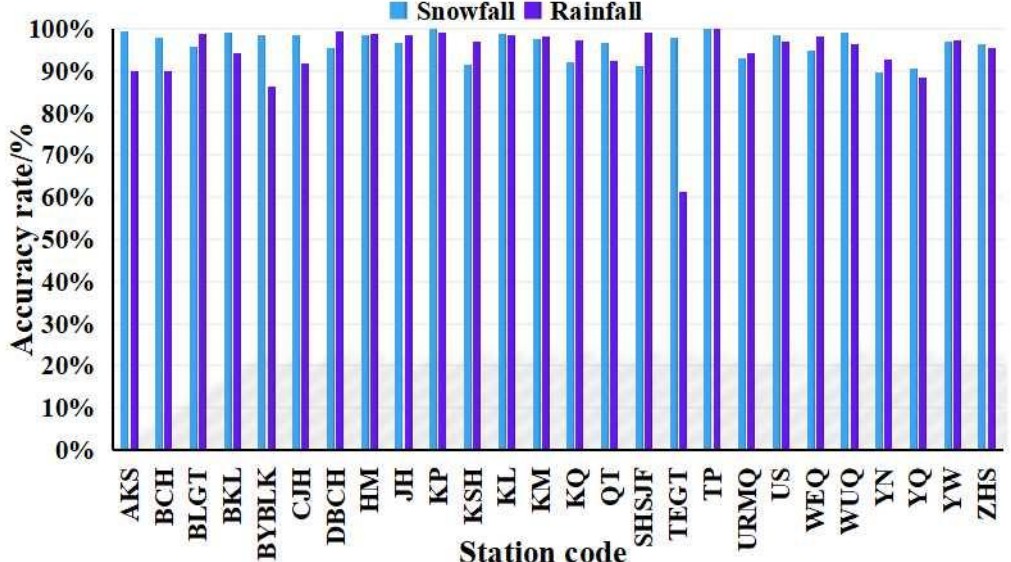

**Figure 2.** Accuracy rate of discriminating snowfall and rainfall using RST value for each meteorological station in the CTMR.



**Table 1.** Descriptive information of meteorological stations selected, including name, code, latitude, longitude, elevation, RST, and duration in the CTMR

| No. | Station name | Station code | Latitude (°N) | Longitude (°E) | Elevation (m) | RST (°C) | Duration |
|-----|-------------|-------------|---------------|----------------|---------------|----------|----------|
| 1 | Aksu | AKS | 80.23 | 41.17 | 1104.73 | 3.00 | 1961-2020 |
| 2 | Baicheng | BCH | 81.90 | 41.78 | 1230.00 | 2.61 | 1961-2017 |
| 3 | Balguntay | BLGT | 86.30 | 42.73 | 1739.47 | 5.35 | 1961-2017 |
| 4 | Barkol | BKL | 93.05 | 43.60 | 1675.37 | 3.40 | 1961-2020 |
| 5 | Bayanbulak | BYBLK | 84.15 | 43.03 | 2459.27 | 1.81 | 1961-2020 |
| 6 | Caijiahu | CJH | 87.53 | 44.20 | 440.97 | 1.05 | 1961-2017 |
| 7 | Dabancheng | DBCH | 88.32 | 43.35 | 1104.77 | 2.98 | 1961-2020 |
| 8 | Hami | HM | 93.52 | 42.82 | 738.20 | 2.91 | 1961-2020 |
| 9 | Jinghe | JH | 82.90 | 44.60 | 320.87 | 1.83 | 1961-2020 |
| 10 | Kalpin | KP | 79.05 | 40.50 | 1162.63 | 3.33 | 1961-2020 |
| 11 | Kashi | KSH | 75.99 | 39.27 | 1291.20 | 3.50 | 1961-2020 |
| 12 | Korla | KL | 86.13 | 41.25 | 932.43 | 3.86 | 1961-2020 |
| 13 | Kumux | KM | 88.22 | 42.23 | 923.47 | 2.03 | 1961-2020 |
| 14 | Kuqa | KQ | 82.97 | 41.73 | 1082.93 | 3.27 | 1961-2017 |
| 15 | Qitai | QT | 89.57 | 44.02 | 794.10 | 1.75 | 1961-2017 |
| 16 | Shisanjianfang | SHSJF | 91.73 | 43.22 | 722.90 | 2.80 | 1961-2017 |
| 17 | Tuergate | TEGT | 75.40 | 40.52 | 3506.40 | 4.70 | 1961-2017 |
| 18 | Turpan | TP | 89.20 | 42.93 | 34.97 | 2.50 | 1961-2020 |
| 19 | Urumqi | URMQ | 87.65 | 43.78 | 935.67 | 1.96 | 1961-2017 |
| 20 | Usu | US | 84.67 | 44.43 | 478.97 | 1.81 | 1961-2017 |
| 21 | Wenquan | WEQ | 81.02 | 44.97 | 1358.60 | 1.60 | 1961-2017 |
| 22 | Wuqia | WUQ | 75.25 | 39.72 | 2176.90 | 3.03 | 1961-2020 |
| 23 | Yining | YN | 81.33 | 43.95 | 663.20 | 1.08 | 1961-2017 |
| 24 | Yanqi | YQ | 86.57 | 42.08 | 1056.60 | 2.35 | 1961-2020 |
| 25 | Yiwu | YW | 94.70 | 43.27 | 1728.60 | 4.15 | 1961-2020 |
| 26 | Zhaosu | ZHS | 81.13 | 43.15 | 1853.40 | 2.44 | 1961-2017 |


## 2.2.2 The modeled future daily air temperature data

The Shared Socioeconomic Paths (SSPs) describe social change in the future in the absence of climate policy intervention within CMIP6 (O'Neill et al., 2016). The SSP126, SSP245, SSP370, and SSP585 scenarios correspond to the anthropogenic radiative forcing stabilized at 2.6 w/m$^2$, 4.5
w/m$^2$, 7.0 w/m$^2$, and 8.5 w/m$^2$ in 2100, respectively (Zhang et al., 2019). Daily air temperature data from 14 models, namely ACCESS-ESM1-5, BCC-CSM2-MR, CanESM5, CESM2-WACCM, CMCC-ESM2, CNRM-CM6-1, CNRM-ESM2-1, INM-CM4-8, INM-CM5-0, IPSL-CM6A-LR, MIROC6, MRI-ESM2-0, NorESM2-LM, and NorESM2-MM were chosen for this paper (see Table 2). The models were capable of providing daily air temperature under the SSP126, SSP245,
SSP370, and SSP585 scenarios spanning 2021 to 2100. Since model resolutions were unsuitable

for regional change and differed from the horizontal resolution of observation data, the horizontal resolution was resampled to 0.25°×0.25°. The multi-model ensemble averaging method and the bi-linear interpolation method were used to interpolate the model data to meteorological stations across the CTMR to mitigate uncertainty among modes and to facilitate better comparison and analysis of model data with the observation data. The delta deviation correction method was employed to correct the deviation between the bi-linear interpolated, multi-models averaged, and the observed daily temperature data (Li et al., 2021). The period, 2021-2100, was divided into 2021-2040, 2041-2070, and 2071-2100, respectively. Finally, the daily 14-model ensemble averaged air temperature under four scenarios during three periods was used to analyze the future change of PSP indicators in the CTMR.

**Table 2.** Basic information of 14 models from the CMIP6

| No. | Model name | Institution | Country/Region | Resolution (lat×lon) |
|-----|-----------|-------------|----------------|----------------------|
| 1 | ACCESS-ESM1-5 | ACCESS | Australia | 1.875°×1.250° |
| 2 | BCC-CSM2-MR | BCC | China | 1.125°×1.125° |
| 3 | CanESM5 | CCCma | Canada | 2.813°×2.813° |
| 4 | CESM2-WACCM | NCAR | America | 2.500°×1.875° |
| 5 | CMCC-ESM2 | CMCC | Italy | 1.406°×1.406° |
| 6 | CNRM-CM6-1 | CNRM-CERFACS | France | 1.406°×1.406° |
| 7 | CNRM-ESM2-1 | CNRM-CERFACS | France | 1.406°×1.406° |
| 8 | INM-CM4-8 | INM | Russia | 2.000°×1.500° |
| 9 | INM-CM5-0 | INM | Russia | 2.000°×1.500° |
| 10 | IPSL-CM6A-LR | IPSL | France | 2.500°×1.259° |
| 11 | MIROC6 | MIROC | Japan | 1.406°×1.406° |
| 12 | MRI-ESM2-0 | MRI | Japan | 1.125°×1.125° |
| 13 | NorESM2-LM | NCC | Norway | 2.500°×1.875° |
| 14 | NorESM2-MM | NCC | Norway | 1.250°×0.900° |

## 2.3 Methods

### 2.3.1 Definition of PSP indicators

The intra-annual distribution of daily mean air temperature for a site can be described using the semi-sinusoidal curve function, which has been previously shown to fit the 10-day air temperature distribution well across Northwestern China (Li et al., 2015). The equations for EPSS, SPSS, and LPSS for a site are as follows:

$$T_i = A_i \sin (w_i t + \varphi_i) \tag{1}$$

$$EPSS_i = (\arcsin(RST/A_i) - \varphi_i)/w_i \tag{2}$$

$$SPSS_i = (\pi - \arcsin(RST/A_i) - \varphi_i)/w_i \tag{3}$$

$$LPSS_{i \to (i+1)} = 365 + EPSS_{i+1} - SPSS_i \tag{4}$$

where $i$ represents the year ($i$=1961, 1962, 1963,…, 2100) and $t$ represents the day of Julian's year (DOY, $t$=1, 2, 3,…, 365 or 366). The semi-sinusoidal curve function simulates the daily mean air temperature time series of the $i$th year. $T_i$, based on shape parameters $A_i$, $\omega_i$, and $\varphi_i$, which are estimated using the nonlinear least-squares method. $RST$ is the rain-snow threshold, and $EPSS_i$ stands for the left intersection point, and $SPSS_i$ for the right within the $i$th year for a site. Similarly, $EPSS_{i+1}$ stands for the left intersection point, and $SPSS_{i+1}$ for the right within the $(i+1)$th year for a site (see Figure 3). In addition, $LPSS_{i \to (i+1)}$ is the horizontal distance from $SPSS_i$ to $EPSS_{i+1}$ and

can be calculated by equation (4). Therefore, EPSS and SPSS are from 1961 to 2017/2020, while the LPSS is from 1962 to 2017/2020 for a site.

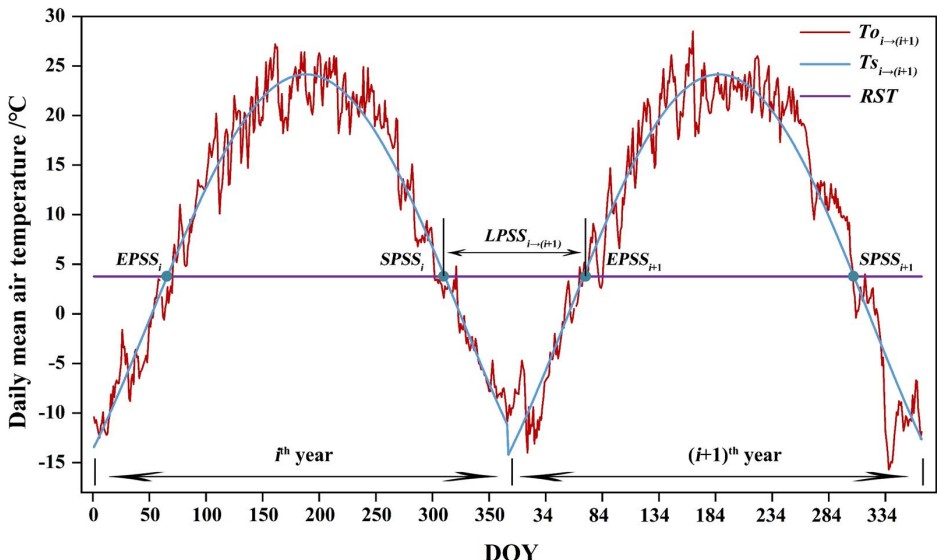

**Figure 3.** The fitting curve, the observed curve of daily mean air temperature within two consecutive years (the $i$th year and the $(i+1)$th year), and three PSP indicators for a location in the CTMR. EPSS, the end of the potential snowfall season; SPSS, the start of the potential snowfall season; LPSS, the length of the potential snowfall season.

### 2.3.2 Mann-Kendall monotonic test method

The non-parametric Mann-Kendall (M-K) test method has the advantages of not requiring data to follow a specific distribution and being less sensitive to outliers and missed values (Kendall, 1990). This method is widely applied for detecting the significance of long-term trends in a time series from hydrology and climatology, such as air temperature, precipitation, and runoff (Li et al., 2011; Li et al., 2022). Detailed information about this method can refer to Kendall (1990) and Li et al. (2011). This paper employed the M-K test method to test the changing trends of PSP indicators throughout the study area.

## 3 Results

### 3.1 General characteristic of PSP indicators past and present

Three PSP indicators, the SPSS, the EPSS, and the LPSS, for each station across the CTMR, were calculated by equation (2)-(4). The SPSS, the EPSS, and the LPSS values differed among 26 stations. The average values of the SPSS, the EPSS, and the LPSS were 307th DOY, 78th DOY, and 136 days, respectively. Figure 4 shows the spatial distribution of the SPSS, EPSS, and LPSS across the CTMR. The SPSS mainly occurred after 301st DOY (approximately October 27th) and ranged from the 256th to 326th DOY (approximately September 12th to November 21st). The EPSS values mainly happened before 61st DOY (March 2nd) and ranged from the 50th to 139th DOY (approximately February 19th to May 18th). The LPSS ranged from 90 to 248 days (approximately three months to eight months) and was shorter than 150 days (approximately five months) in most of the region. The SPSS was earlier, the EPSS was later, and the LPSS was longer in the north and center than in the south. In some areas like BYBLK and TEGT, where the LPSS was longer than 200 days (approximately seven months). On average, the potential snowfall started November 2nd,

ended about March 18th, and lasted about 136 days across the CTMR.

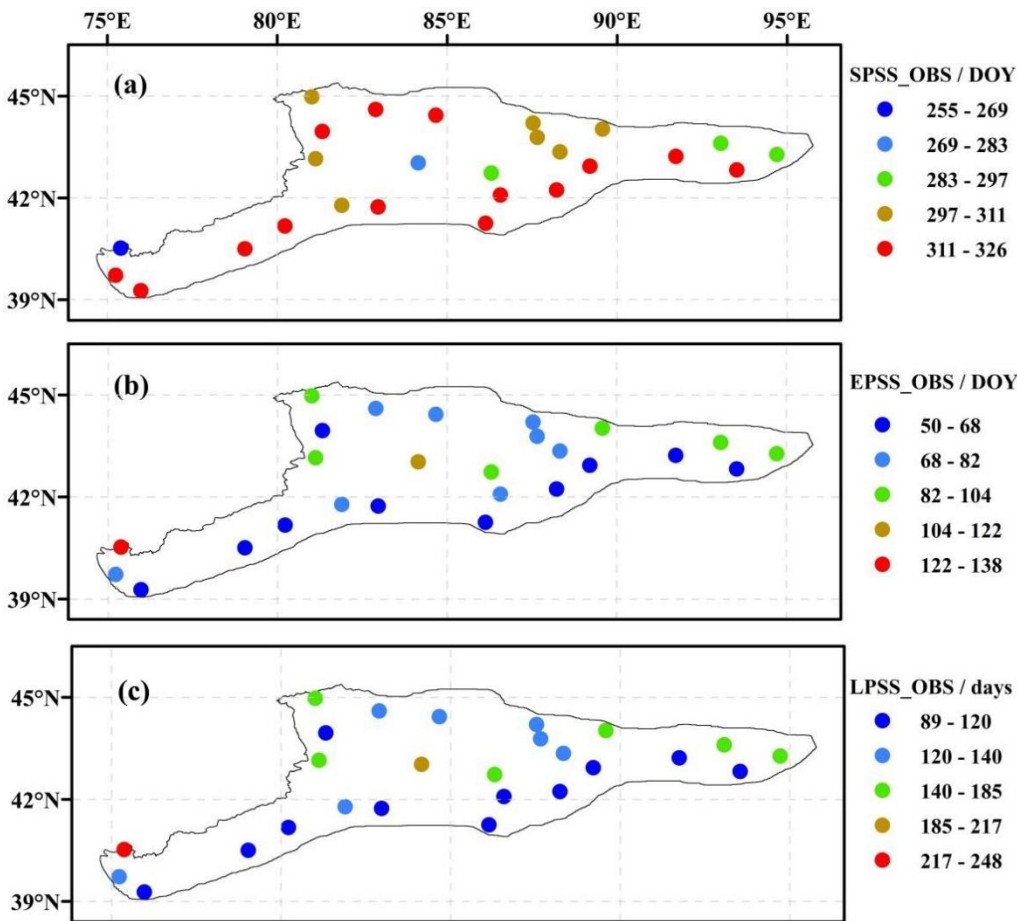

**Figure 4.** Spatial distribution of the average value of the SPSS (a), the EPSS (b), and the LPSS (c) in the CTMR.

### 3.2 Temporal variation of the PSP indicators in the past and present

All three indicators of the PSP exhibited a noticeable trend during the study period. The SPSS and EPSS both had a significant increasing and decreasing trend, respectively, with a 0.05 confidence level over the last six decades (see Figure 5a). Similar to the EPSS, a statistically significant decreasing trend was observed for the LPSS from 1962 to 2017/2020 (see Figure 5b). The slopes of SPSS, EPSS, and LPSS were 1.2 days/10a, -1.6 days/10a, and -2.8 days/10a, respectively. This means that the potential snowfall season had a delayed start rate of 1.2 days per decade, an earlier ending rate of 1.6 days per decade, and a reduction of 2.8 days per decade in the duration of the potential snowfall. Moreover, the rate of advancement for the EPSS was higher than that of the SPSS delay.

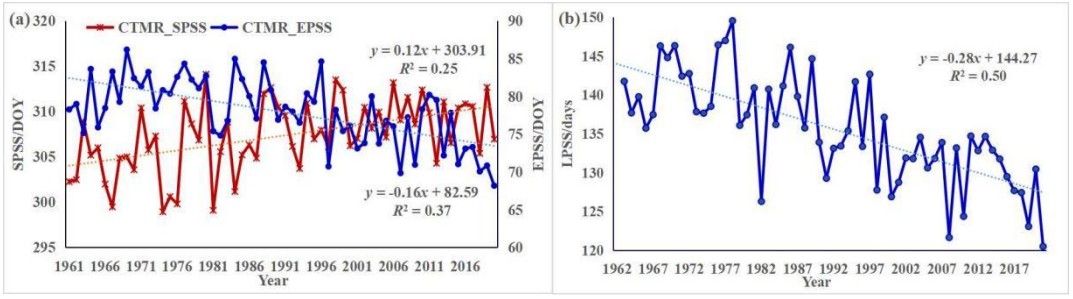

**Figure 5.** Annual time series and their trends of the SPSS and EPSS (a) during 1961-2020 and the LPSS (b) during

1962-2020 in the CTMR.

**3.3 Spatial pattern of PSP indicators in the past and present**

During 1961-2017/2020, a majority of the studied region experienced an upward trend in SPSS
and a downward trend in both EPSS and LPSS, respectively. This implies that potential snowfall
mostly started later, ended earlier and lasted for shorter duration. Five stations, namely DBCH,
HM, KL, KQ, and KP, did not exhibit any significant changing trends for the SPSS. Additionally,
BYBLK, KQ, and TEGT showed no significant changes in EPSS. LPSS did not exhibit a
significant changing trend in KQ and TEGT due to the non-significant changing trend of SPSS or
EPSS. SPSS, which increased in most of the area, caused the potential snowfall season to start
later by approximately 2-13 days. Conversely, EPSS decreased across the area, causing the
potential snowfall season to end earlier by about 1-13 days. Consequently, LPSS experienced a
reduction of approximately 3-26 days across the area (see Figure 6). This reduction in potential
snowfall days may cause a decrease in the annual total snowfall (Räisänen, 2016).

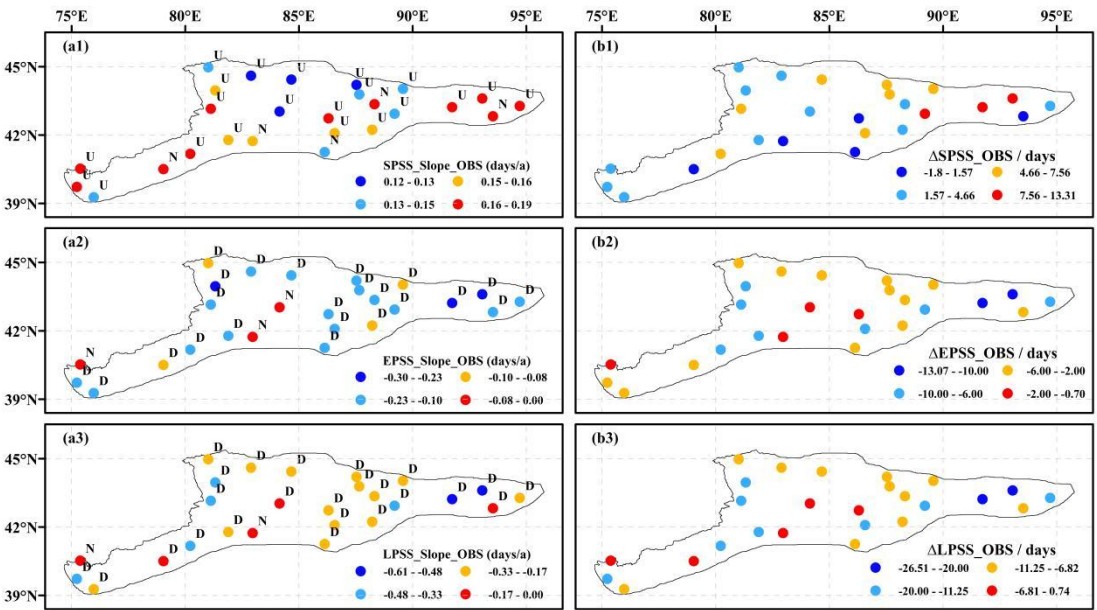

**Figure 6.** Changing trends and slopes of the SPSS (a1), EPSS (b1), and LPSS (c1) and their amplitude of variation
(a2-c2) across the CTMR during 1961/1962-2017/2020. D for the significant downward trend at the 0.05
significance level, N for no trend, and U for the significant upward trend at the 0.05 significance level.

**3.4 Temporal tendency of PSP indicators in the future**

Figure 7 shows the time series of three PSP indicators in the CTMR from 2021 to 2100 and their
average values during three periods (2021-2040, 2041-2070, and 2071-2100) under four different
scenarios.

The SPSS is projected to increase under all four scenarios from 2021 to 2100, and the rate of
increase is steeper under the higher emission scenario compared to that with lower emission
scenario (see Figure 7a1). The average values of the SPSS are expected to increase from
2021-2040 to 2071-2100 under all four scenarios, with the SSP585 scenario showing the steepest
rise (see Figure 7a2). Under the SSP585 scenario, the average value of the SPSS is projected to
reach 326th DOY during 2071-2100, indicating a potential snowfall season starting on November

 21st in 2100s.

In contrast, the EPSS will initially remain constant and then increase under the SSP126 scenario, while decreasing trends in the EPSS are expected under the other three scenarios. Values of the EPSS are projected to decrease slightly from the SSP245 scenario to the SSP585 scenario (see Figure 7b1). Average values of the EPSS will change less during the first two periods and increase slightly during 2071-2100 under the SSP126 scenario. Nevertheless, average values of the EPSS drop steeper and steeper in the three future periods under the SSP245 scenario to the SSP585 scenario (see Figure 7b2). Under the SSP585 scenario, the average value of the EPSS is expected to reach 65th DOY during 2071-2100, implying that the potential snowfall season ending before March 5th in 2100s across the CTMR.

Furthermore, the LPSS is projected to decrease in the coming 80 years under all four scenarios, with a steeper downward slope from the SSP126 scenario to the SSP585 scenario (see Figure 7c1). Average values of the LPSS are expected to decrease from 2021-2040 to 2071-2100 under all four scenarios. The drop will become more pronounced from the SSP126 scenario to the SSP585 scenario. Under the SSP585 scenario, the potential snowfall season is expected to be shortened by up to 48 days, lasting only 88 days in 2071-2100 (see Figure 7c2).

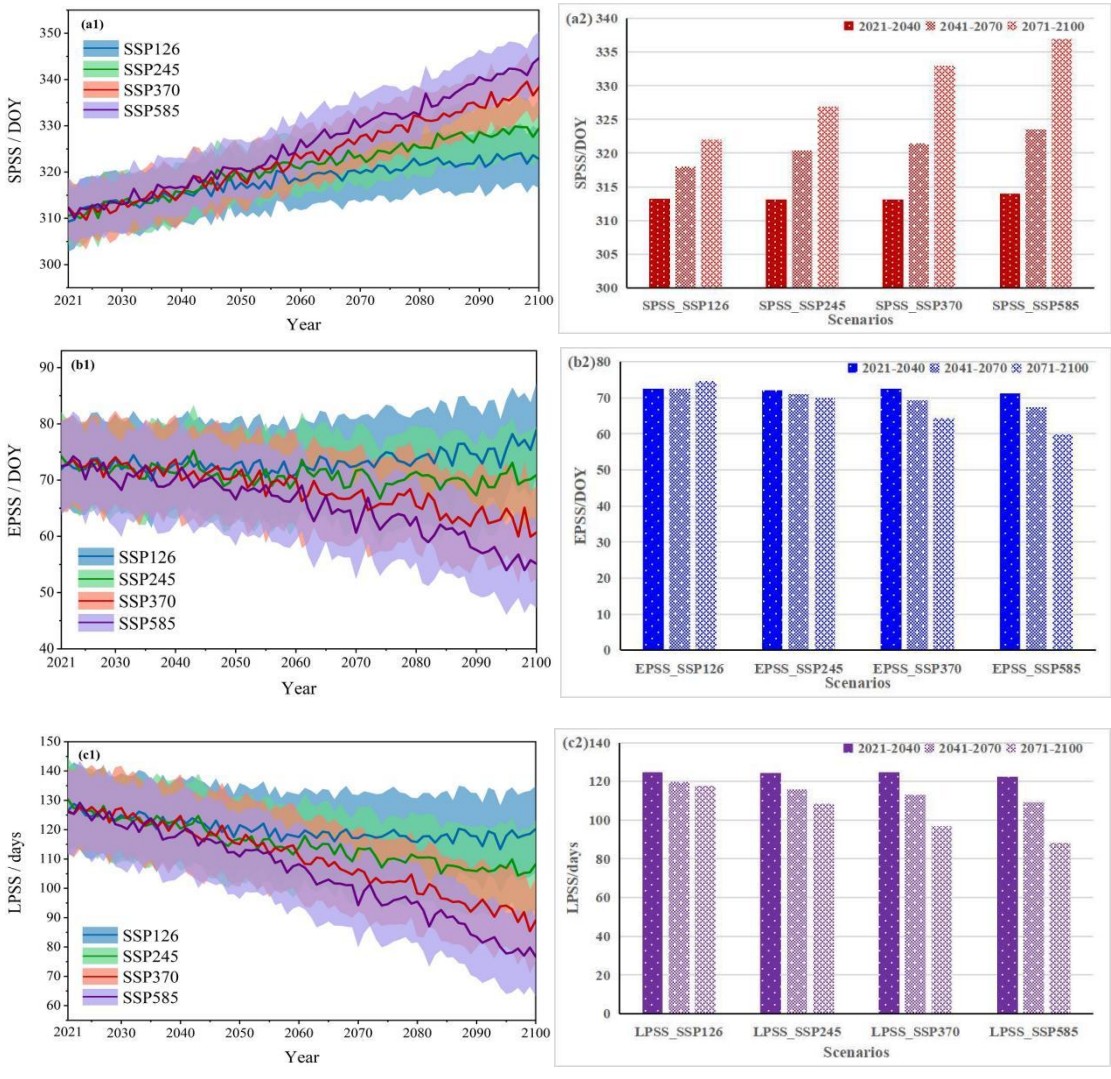

**Figure 7.** Time series of the SPSS (a1), the EPSS (b1), and the LPSS (c1) during 2021-2100 and their average values (a2-c2) during three periods (2021-2040, 2041-2070, and 2071-2100) under all four scenarios (SSP126,

SSP245, SSP370, and SSP585) in the CTMR.

**3.5 Spatial image of PSP indicators in the future**

Figure 8 shows the spatial distribution of the SPSS, EPSS, and LPSS under four scenarios across the CTMR. Values of the SPSS are expected to fall between 266th and 337th DOY (approximately September 22nd to December 1st) under the SSP126 scenario, between 269th and 340th DOY (approximately September 25th to December 4th) under the SSP245 scenario, between 273rd and 342nd DOY (approximately September 30th to December 6th) under the SSP370 scenario, and between 278th and 346th DOY (approximately October 4th to December 10th) under the SSP585 scenario, respectively. The EPSS is expected to happen between 46th and 140th DOY (approximately February 15th to May 19th) under the SSP126 scenario, between 44th and 138th DOY (approximately February 13th to May 17th) under the SSP245 scenario, between 42nd and 135th DOY (February 11th to May 14th) under the SSP370 scenario, and between 40th and 130th DOY (approximately February 9th to May 9th) under the SSP585 scenario, respectively. And the LPSS is expected to last between 78 and 239 days (approximately 2 months and 18 days to 8 months) under the SSP126 scenario, between 73 and 234 days (approximately 2 months and 13 days to 7 months and 24 days) under the SSP245 scenario, between 69 and 227 days (approximately 2 months and 9 days to 7 months and 17 days) under the SSP370 scenario, and between 64 and 217 days (approximately 2 months and 4 days to 7 months and 7 days) under the SSP585 scenario, respectively. As in the past and present, the SPSS will occur earlier, the EPSS will occur later, and the LPSS will last longer in the north and center compared to the south, particularly in BYBLK and TEGT, where the LPSS will last for more than 6 months under all four scenarios.

In the coming 80 years (2021-2100), the SPSS in the entire region is expected to have significant upward trends with gradually increasing rates under the SSP126 scenario to the SSP585 scenario, indicating that the potential snowfall season will start much later under higher emission scenario. The amplitude of variation in the SPSS will increase with the enhancement of scenarios from the SSP126 to the SSP585. The SPSS under the SSP585 scenario is expected to be delayed for 24-41 days. The west and southwest parts will have a faster postponement of the SPSS than the center, and it will delay the starting time of the potential snowfall season more in the west and southwest under all four scenarios (see Figure 9).

Under the SSP245, SSP370, and SSP585 scenarios, the EPSS displays significant downward trends with gradually increasing rates (slopes) in most of the region. Conversely, under the SSP126 scenario, the EPSS will exhibit significant upward trends ranging from 0.1 day/10a to 1.1 day/10a across all the region. Therefore, the amplitude of variation in the EPSS will also be non-uniform under four future scenarios. Under the SSP126 scenario, the end of the potential snowfall season will be delayed for 1-7 days across all the region, which will be advanced by 9-23 days under the SSP585 scenario. In the west and southwest of the CTMR, the EPSS will decrease more rapidly, and the ending time will advance further under the SSP245, SSP370, and SSP585 scenarios (see Figure 10).

Consequently, the LPSS exhibits notable negative trends with progressively increasing slopes from the SSP126 scenario to the SSP585 scenario. It will contract the duration of the potential snowfall season by 5-10 days, 12-26 days, 24-45 days, and 33-61 days under the SSP126, SSP245, SSP370, and SSP585 scenarios, respectively. In the west and southwest of the CTMR, the LPSS

 will be truncated even further under all four scenarios (see Figure 11).

In summary, while SPSS will increase significantly with the scenario increase from the SSP126 to the SSP585, both the EPSS and LPSS will undergo substantial declines. By the end of this century, the start of the potential snowfall season is projected to be delayed by 24-41 days under the SSP585 scenario, while the end of the potential snowfall season is projected to be with a delay of 1-7 days under the SSP126 scenario and an advancement of 9-23 days in advance under the SSP585 scenario, respectively. Consequently, the SSP585 scenario will shorten the length of the potential snowfall season by 33-61 days. Spatially, the western and southwestern parts of the CTMR will experience more rapid changes in the starting time, ending time, and length of the potential snowfall season compared to other parts.

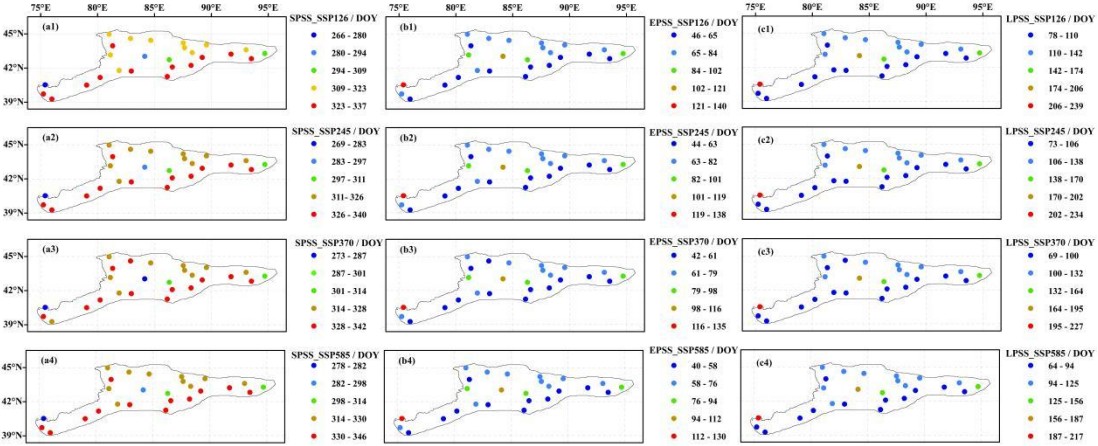

**Figure 8.** Spatial distribution of the average value of the SPSS (a1-a4), EPSS (b1-b4), and LPSS (c1-c4) under four different scenarios (SSP126, SSP245, SSP370, and SSP585) across the CTMR during 2021-2100.

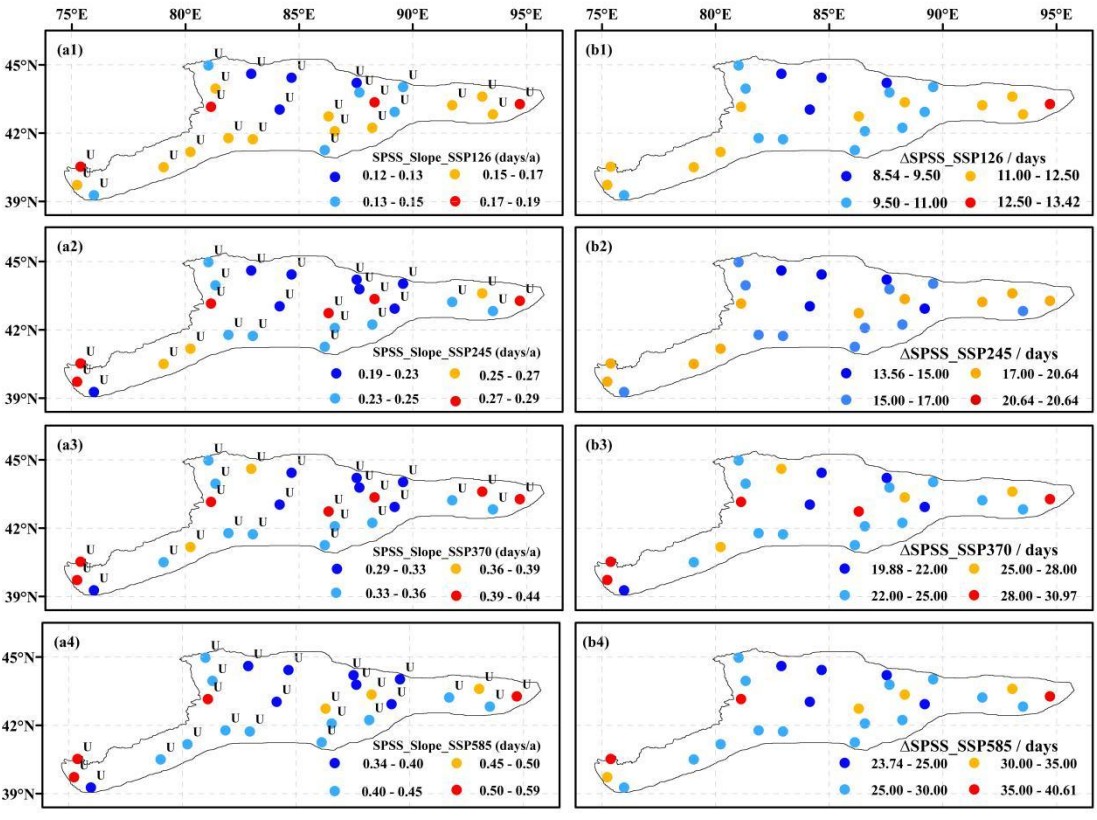

**Figure 9.** Changing trends and slopes of the SPSS under the SSP126 (a1), SSP245 (a2), SSP370 (a3), and SSP585 (a4) scenarios and its amplification (b1-b4) across the CTMR during 2021-2100. D for the significant downward trend at the 0.05 significance level, N for no trend, and U for the significant upward trend at the 0.05 significance level.

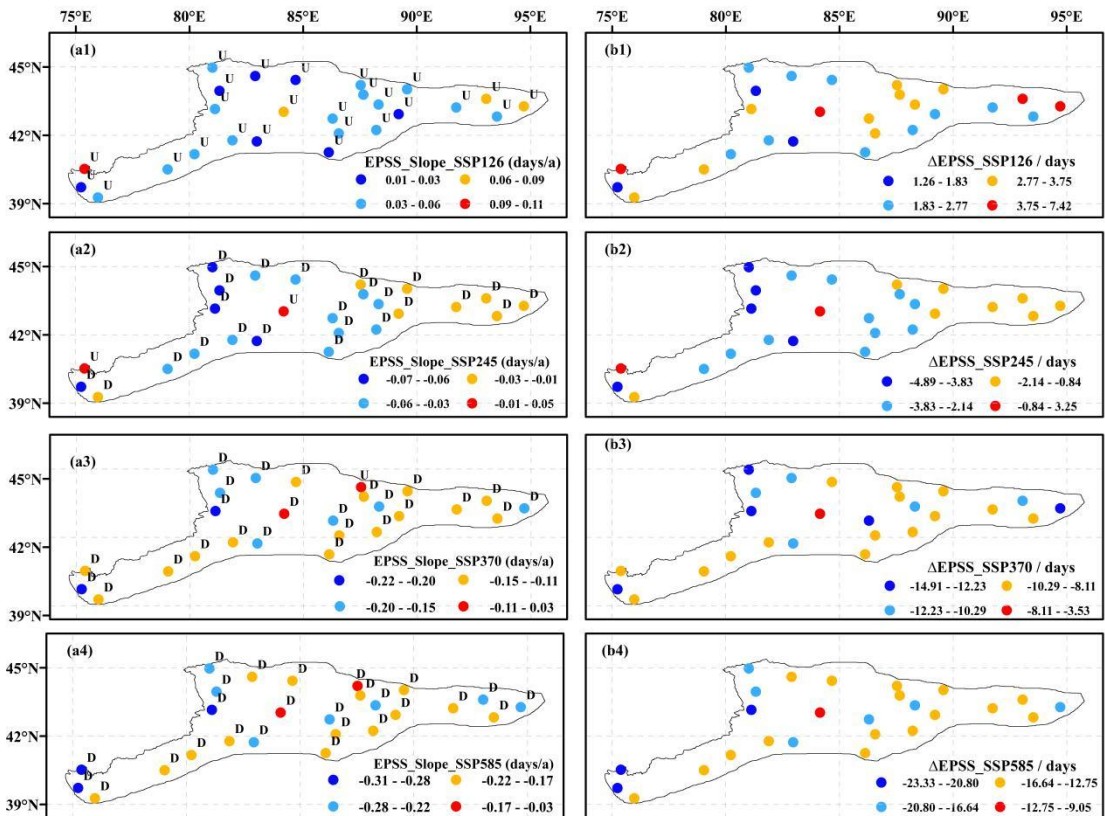

**Figure 10.** Changing trends and slopes of the EPSS under the SSP126 (a1), SSP245 (a2), SSP370 (a3), and SSP585 (a4) scenarios and its amplification (b1-b4) across the CTMR during 2021-2100. D for the significant downward trend at the 0.05 significance level, N for no trend, and U for the significant upward trend at the 0.05 significance level.

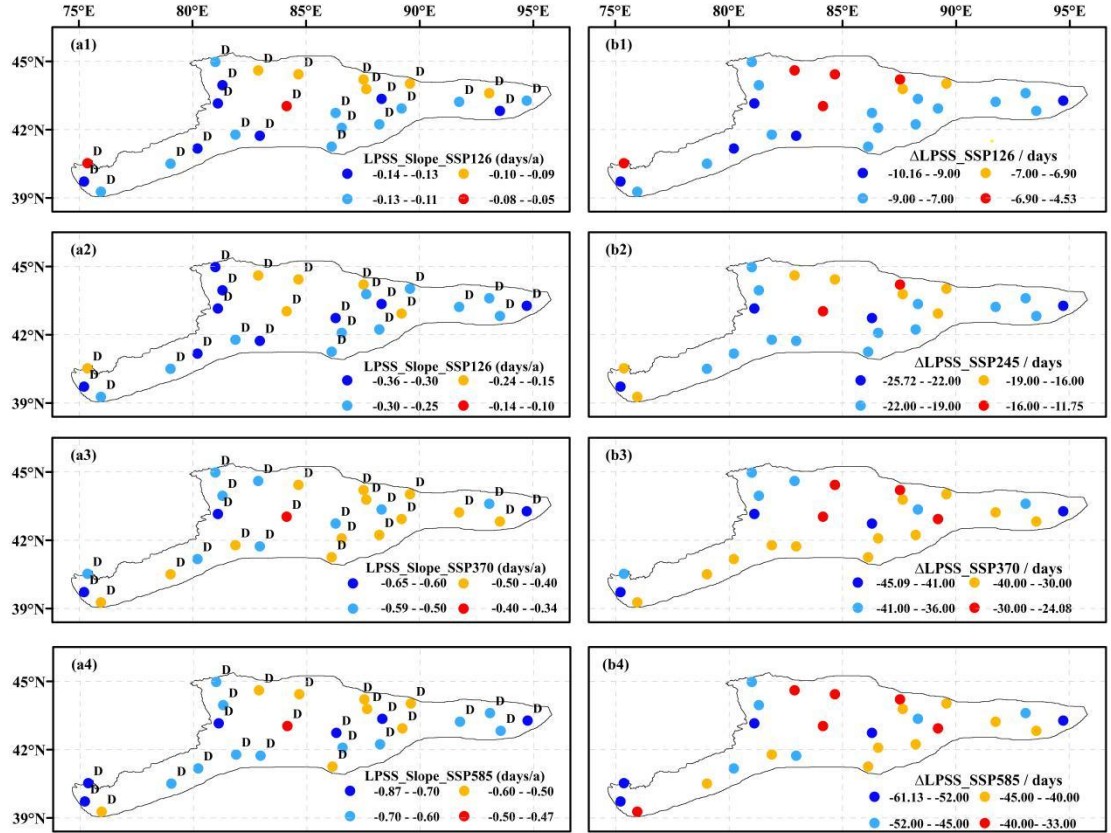

**Figure 11.** Changing trends and slopes of the LPSS under the SSP126 (a1), SSP245 (a2), SSP370 (a3), and SSP585 (a4) scenarios and its amplification (b1-b4) across the CTMR during 2021-2100. D for the significant downward trend at the 0.05 significance level, N for no trend, and U for the significant upward trend at the 0.05 significance level.

## 4 Discussion

### 4.1 Performance of PSP indicators

Much previous work focused on variability, trends, spatial-temporal pattern, extremes, and events of snowfall over the snow-dominated regions worldwide (Baijnath-Rodino et al., 2018; Nazzareno et al., 2019; Bai et al., 2019; Takahashi et al., 2021; Lin and Chen, 2022). However, less attention was paid to the change in phenology of potential snowfall. To address this knowledge gap, we skillfully presented three indicators, namely, the SPSS, EPSS, and LPSS, based on the semi-sinusoidal curve of daily air temperature combined with various rain-snow thresholds. The SPSS and EPSS represent the start and end of the potential snowfall season within a given year, respectively, while the SPSS denotes the difference between the EPSS in the second year and the SPSS in that same year. Our work showed the potential snowfall season mainly occurred from October 30[th] to March 20[th] during 1961-1979 across the CTMR. In comparison, the observed snowfall season was from October 20[th] to March 17[th] during the same period. The start/end dates of potential snowfall season calculated were significantly correlated with the observed ones at the 0.01 significance level, with correlation coefficients of 0.56 and 0.88, respectively (Figures omitted). Moreover, Tian et al. (2020) exhibited snowfall season in the CTMR was from November to March. Thus, the consistency between the observed and the potential snowfall season was high, and the potential snowfall season could match or cover the observed one. These

findings indicate that the SPSS, EPSS, and LPSS accurately reproduce the phenology of potential snowfall across the CTMR and have the potential to be expanded to other snow-dominated regions worldwide.

**4.2 Temporal heterogeneity**

The PSP indicators showed a relative continuity in their trends from past to future. Both the historical (1961-2020) and future (2021-2100) periods revealed a consistent upward or downward trend in the SPSS and LPSS time series, despite the observed SPSS and LPSS displaying greater fluctuations (see Figure 12a and 12c). However, the EPSS showed an exception in its changing trend. The changing trends of the observed EPSS were consistent with those of the projected EPSS under SSP245, SSP370, and SSP585 scenarios, but were opposite to that of the projected EPSS under SSP126 scenario (see Figure 12b). Furthermore, unlike the historical period, the rate of postponing SPSS was faster than that of advancing EPSS under all four scenarios, which may be due to seasonal diversity in the warming rate of air temperature. The amplitude of air temperature variation under all four scenarios will be larger in autumn and winter compared to the other two seasons (see Figure 13). The higher emission scenario will result in a higher rate of climate warming, leading to the postponing SPSS and advancing EPSS, ultimately accelerating the reduction of LPSS from the historical period to the SSP585 scenarios in the future (see Table 3).

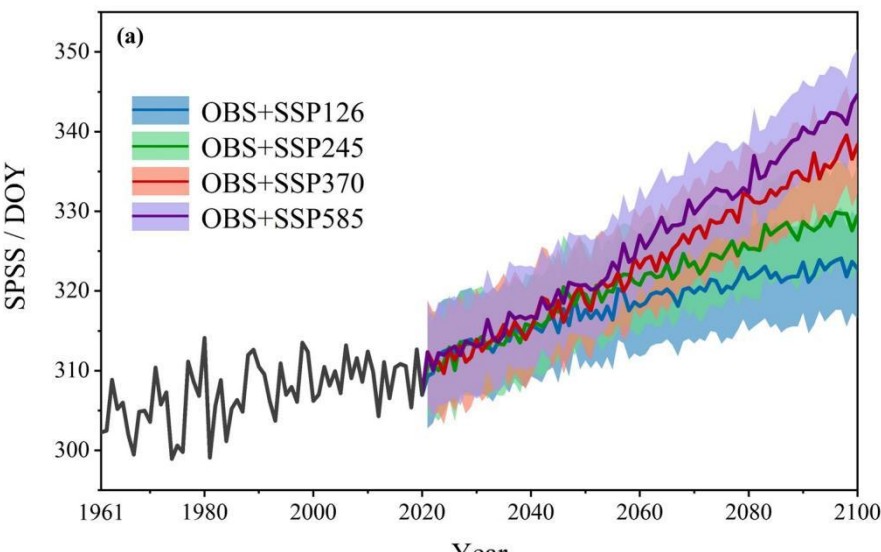

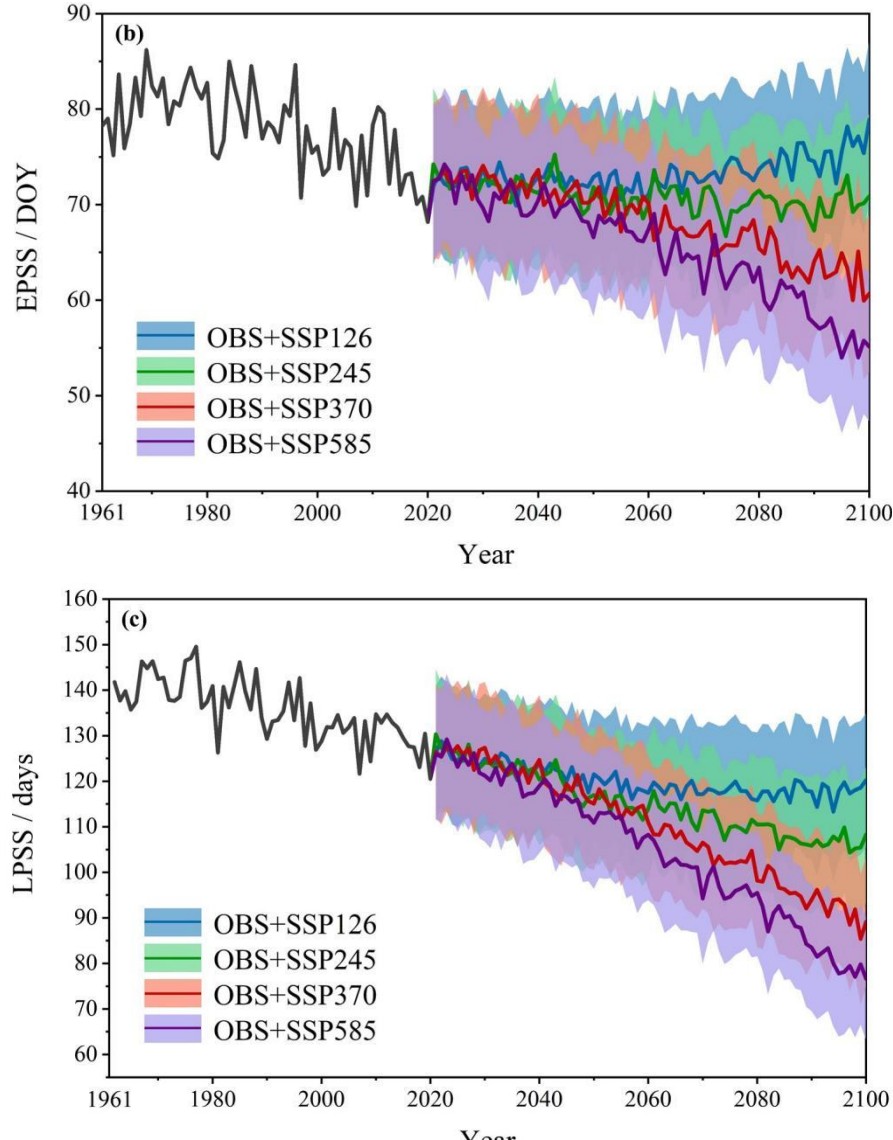

**Figure 12.** Time series of the SPSS (a), EPSS (b), and LPSS (c) from 1961 to 2100 in the CTMR. OBS means time series based on observation data during 1961-2020; SSP126, SSP245, SSP370, and SSP585 refer to time series based on CMIP6 data during 2021-2100.

**Table 3.** The changing rate in PSP indicators in the CTMR

|  | SPSS | EPSS | LPSS |
| --- | --- | --- | --- |
| OBS | 1.16 days/10a | -1.55 days/10a | -2.61 days/10a |
| SSP126 | 1.52 days/10a | 0.40 days/10a | -1.13 days/10a |
| SSP245 | 2.44 days/10a | -0.34 days/10a | -2.83 days/10a |
| SSP370 | 3.62 days/10a | -1.52 days/10a | -5.00 days/10a |
| SSP585 | 4.23 days/10a | -2.22 days/10a | -6.40 days/10a |

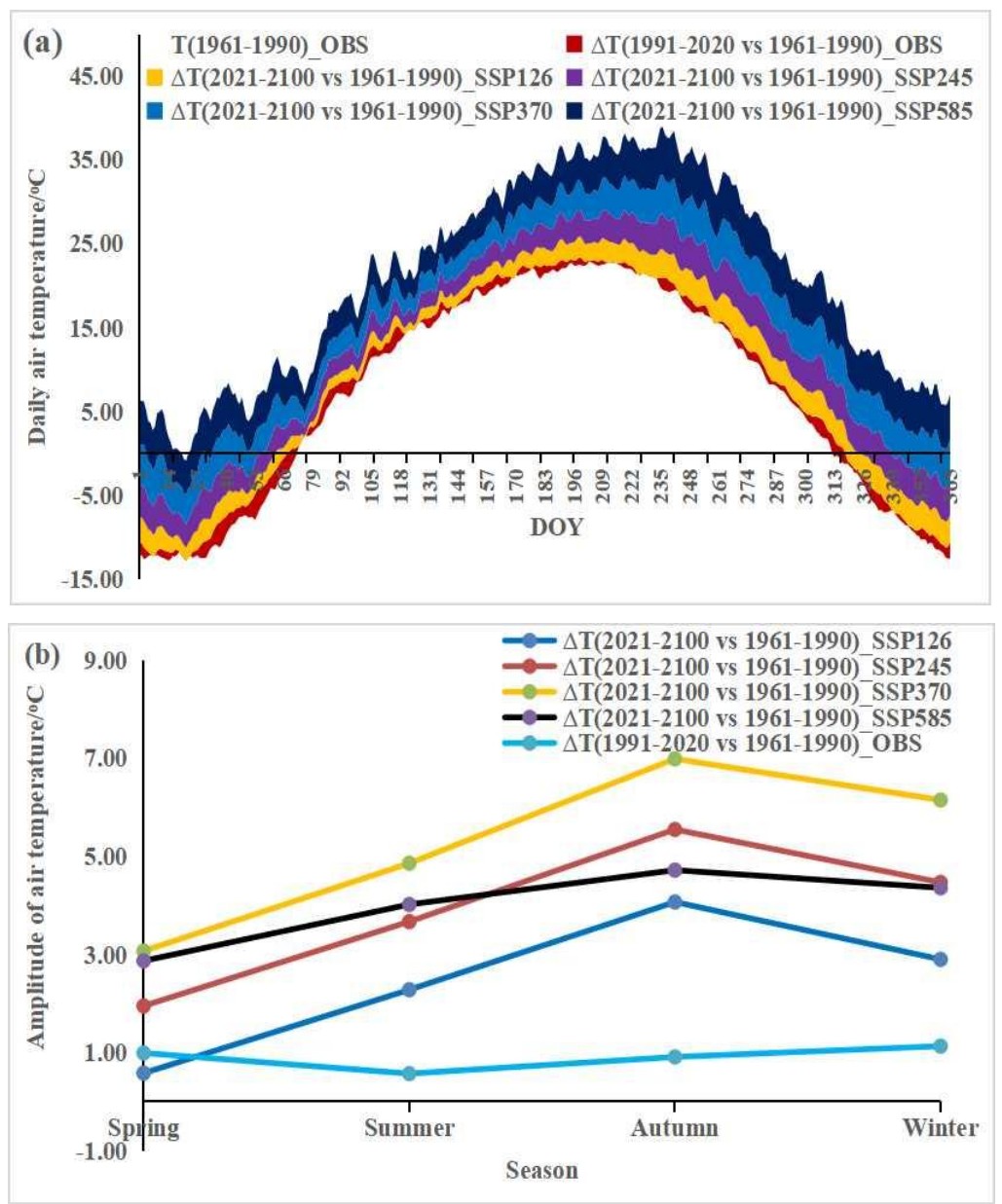

**Figure 13.** The intra-annual amplitude of daily air temperature variation (a) and its seasonal diversity (b) in the CTMR. T(1961-1990)_OBS is the observed mean daily air temperature during 1961-1990; ∆T(1991-2020 vs. 1961-1990)_OBS is the amplitude of air temperature variation during 1991-2020 compared to that during 1961-1990; ∆T(2021-2100 vs. 1961-1990)_SSP126 means the amplitude of air temperature variation during

2021-2100 under the SSP126 scenario compared to that during 1961-1990; the same to ∆T(2021-2100 vs. 1961-1990)_SSP245, ∆T(2021-2100 vs. 1961-1990)_SSP370, and ∆T(2021-2100 vs. 1961-1990)_SSP585.

## 4.3 Uncertainty

The distribution of meteorological stations in the CTMR is uneven, particularly in high elevation areas, and available data is limited to observations during 1950s-1979, which may compromise the

accuracy of the three PSP indicators. Although CMIP6 data could capture surface air temperature trends well (Fan et al., 2020), it can not replicate seasonal diversity with the same level of accuracy. The correlation between the observed and the simulated air temperature from CMIP6 during 1961-2020 ranged from 0.93 to 0.98 (see Figure 14a). In spring, air temperature was underestimated by 0.26 °C, while it was slightly overestimated in the other three seasons (see

Figure 14b). This could lead to the overestimation of both the SPSS and EPSS under the four different scenarios. Additionally, the methodology used in this study, which is based on a semi-sinusoidal curve of daily air temperature combined with various rain-snow thresholds, is subject to certain limitations due to its failure to account for the impacts of wind, humidity, and other meteorological factors on precipitation phase, as well as the influence of factors such as elevation, latitude, and topography on the rain-snow threshold (Li et al., 2020a; Li et al., 2020b). Furthermore, uncertainty may arise from the selection of data, and the use of only 14 models may not completely eliminate modeling errors. Further research could benefit from incorporating data from CMIP6 models or others and improving the methodology.

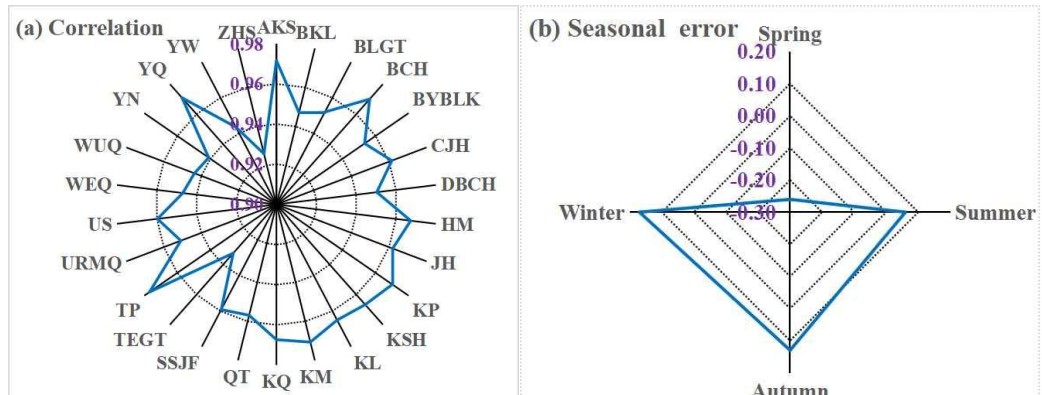

**Figure 14.** Correlation between the observed and the simulated air temperature from CMIP6 at meteorological station scale (a) and its seasonal mean error (b) across the CTMR.

## 5 Conclusions

In this study, we initially defined three indicators of the potential snowfall penology (the SPSS, EPSS, and LPSS) and analyzed their spatial-temporal variation in the past, present, and future over the CTMR. The indicators (the SPSS, EPSS, and LPSS) were found to effectively reproduce the feature of potential snowfall phenology and could be recommended for use in other snow-dominated regions worldwide.

The penology indicators for potential snowfall showed a relative continuity from the past to the future. During 1961-2017/2020, potential snowfall season started on approximately November 2nd, ended on approximately March 18th, and lasted around 136 days across the CTMR on average. However, a significant delay in the starting time, advancement in the ending time, and reduction in the duration of the potential snowfall season was observed. The potential snowfall season started later for 1-13 days at a rate of 1.2 days per decade while the ending time was brought forward by 2-13 days at a rate of 1.6 days per decade, leading to a shorter potential snowfall duration of 3-26 days at the rate of 2.8 days per decade across all the region. The north and center experienced earlier starting time, later ending time, and longer potential snowfall season compared to the south, particularly in BYBLK and TEGT, where the potential snowfall season lasted for over 7 months.

Over the next 80 years (2021-2100), the starting time, ending time, and duration of the potential snowfall season are expected to vary under four scenarios. The higher emission scenario will lead to much later starting time, earlier ending time, and a shorter potential snowfall season due to the higher rate of climate warming. The starting time will be postponed with gradually upward slopes under the scenarios from SSP126 to SSP585. Under the SSP585 scenario, the potential snowfall season will be from about October 4th to December 9th. Ending time will change less during

2021-2070 and be postponed slightly during 2071-2100 under the SSP126 scenario, showing opposite changing trends from the SSP245 scenario to the SSP585 scenario. In contrast, the ending time under the SSP585 scenario will range from February 9th to May 9th across the CTMR. The length will gradually decrease from the SSP126 scenario to the SSP585 scenario in the coming 80 years. The duration of the potential snowfall season will be reduced by up to 61 days by the end of this century under the SSP585 scenario. In the west and east of the CTMR, the length of the potential snowfall season will be cut down by more days due to more delayed starting time and advanced ending time under all four scenarios.

Spatial and temporal heterogeneity existed across the CTMR for the seasonal diversity of warming in the past and present. Uncertainties regarding future projections remain due to the quality and quantity of data used. It is important to note that underestimating air temperature in spring may result in overestimation of the EPSS, while overestimating air temperature in autumn and winter may lead to overestimation of the SPSS when using CMIP6 data for the four future scenarios (2021-2100). Furthermore, the methodology employed in this study has limitations in terms of considering the effects of elevation, latitude, topography, and meteorological factors on precipitation phase and the rain-snow threshold. Further research could focus on enhancing the methodology, gathering multivariate data, and extending the application of these indicators to a broader region.

*Code/Data Availability.* Data are available upon reasonable request to the corresponding author.

*Author contributions.* Xuemei Li wrote and organized the main text; Xinyu Liu and Xu Zhang processed data about air temperature and precipitation phase; Kaixin Zhao provided figures; Lanhai Li offered further insight, comments, and editorial suggestions.

*Competing interests.* The authors declare that they have no conflict of interest.

*Acknowledgements.* This work was financially supported by the National Natural Sciences Foundation of China (42261026, 41971094, and 42161025), Gansu Science and Technology Research Project (22ZD6FA005), and Higher Education Innovation Foundation of Education Department of Gansu Province (2022A-041).

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
