# Peer review of "Change in the potential snowfall phenology: past, present, and future in the Chinese Tianshan mountainous region, Central Asia"

_The Cryosphere, 2022_

## Author Comment (AC1)

**Referee#1**

Global warming speeds up the solid-liquid water cycle and change snowfall phenology. It was found that postponed snowfall occurrence and advanced snowfall ending took place in the Eurasian continent. Potential snowfall phenology can identify the possible onset, end, and duration of snowfall. To describe the characteristics of the potential snowfall phenology, this manuscript proposed three indicators, the start of potential snowfall season (SPSS), the end of potential snowfall season (EPSS), and the length of potential snowfall season (LPSS). Spatial-temporal variations of those three PSP indicators past, present, and future across the Chinese Tianshan mountainous region (CTMR) were explored. The research is sound and thorough. It provides a new direction to understand the potential snowfall phenology in the alpine region. Therefore, I recommend minor revision of this manuscript.

Reply: Thanks for your recognition of our work. Your specific comments could help us to improve our manuscript greatly. We will revise our manuscript seriously according to your specific comments.

Some specific comments and replies are as follows:

1 Lines 35-37: please check the number of days cut down for the length, 63 or 64?

Description on the modification: Thank you very much for your advice. We have checked the number as suggested.

2 Lines 41-43: It suggested that "The results indicate that with constant snowfall intensity, annual total snowfall will decrease, including amount and frequency, ......."

Description on the modification: Thank you very much for your advice. We have revised the sentence as suggested.

3 Lines 50-105: The status, shortcomings and need of potential snowfall phenology studies is poorly described in the introduction.

Description on the modification: we will show more details about status, shortcomings, need, or significance of potential snowfall phenology in the revised manuscript.

4 Lines 95-100: The research from Jennings et al. (2018) only illustrated that RST was spatially heterogeneous and does not show that different methods of precipitation pattern separation yield different RST. Please check it.

Description on the modification: we will check it and revise the related sentences.

5 Lines 128: The sentence "The frequency of rainfall increases while that of snowfall decreases. Besides, precipitation shifting from solid to liquid is obvious" is not clear enough, please check it.

Description on the modification: Thank you very much for your advice. We have revised the sentence as suggested.

6 In Figure 1: It is recommended that a general overview map could be added to Figure 1 to help the reader quickly identify the location of the study area.

Description on the modification: Thank you very much for your good suggestion. We will revise Figure 1 as suggested.

7 In Table 1: It is suggested that a column could be added to Table 1 to indicate the duration of the data.

Description on the modification: Thank you very much for your advice. We have added the duration of the data in Table 1 as suggested.

8 Line 230-232: It is recommended to use "advance / delay" instead of using "smaller or larger" to describe the change of LPSS and EPSS as much as possible.

Description on the modification: Thank you very much for your advice. We have revised the sentence as suggested.

9 Line 241-242: please check the slope of LPSS.

Description on the modification: Thank you very much for your advice. We checked the slope of LPSS. It should be -2.7 days/10a. We changed the text and replace the corresponding figure.

10 Line 371: More should be added here on the comparison of potential snowfall phenology with observed snowfall phenology.

Description on the modification: Thank you very much for your good suggestion. We will collect more observed data to validate the performance of potential snowfall phenology.

11 line 377: "4.2 Spatial and temporal heterogeneity" should be changed to "4.2 Temporal heterogeneity".

Description on the modification: Thank you very much for your advice. We have revised the sentence as suggested.

---

## Author Comment (AC2)

**Referee#2**

General comments:

The manuscript by Li et al. mainly investigated the changes in potential snowfall phenology for the past, present, and future periods in Tianshan, China. They defined three potential snowfall season metrics based on temperature data. Although the definition of potential snowfall phenology is interesting, the hydrological meaning of these metrics is questionable. The methodology of the study was not clearly described. The results lack a strict validation. The credibility of the study needs to be further improved.

Reply: Thanks for your good comments and suggestion, in particular, for your recognition of defining potential snowfall phenology. Snowfall is a solid phase of precipitation affects the ecological environment and hydrological processes in mountainous areas and an important water resource (Barnett et al., 2005; Jonas et al., 2008; Bai et al., 2019). One of the most prominent impacts of climate warming has been a shift from snow to rain in temperate and cold regions across the globe (Knowlesetal., 2006; Trenberth, 2011, Jennings and Molotch, 2019). In this context, the snowfall season will inevitably be cut down. Inspired by definitions of vegetation phenology and snow cover phenology (Lu et al., 2006; Piao et al., 2008; Da Silva et al., 2015; Dahlin et al., 2015; Thackeray et al. 2016; Wang et al. 2016; Li et al., 2017; Zhang et al., 2022), we defined potential snowfall phenology and employed three indicators, start of potential snowfall season (SPSS), the end of potential snowfall season (EPSS), and the length of potential snowfall season (LPSS), to identify the possible onset, end, and duration of snowfall. The advancing or delaying of SPSS means potential snowfall comes earlier or later in late-autumn or early-winter, which affect accumulation and storing of solid water resource as snow cover. Likewise, the advancing or delaying of EPSS indicates potential snowfall ends earlier or later in late-winter or early-spring, which is likely to influence the snow-melting, snow albedo, and runoff yield and concentration in mountainous areas such as the Chinese Tianshan mountainous region (CTMR). Above all, the potential snowfall phenology has hydrological significance in snow-dominated region. The the motivation and methodology of the study were not clearly described, we will revise it in the revised manuscript. Besides, in **4.1 Performance of PSP indicators**, our work showed the potential snowfall season was able to cover the observed one, which partially validated our results. we will add more detailed validation in the revised manuscript. Related references are as followed:

Da, Silva, A., Valcu, M., Kempenaers, B.: Light pollution alters the phenology of dawn and dusk singing in common European songbirds, Phil. Trans. R. Soc. B., 370, 1-9, https://doi.org/10.1098/rstb.2014.0126, 2015.
Dahlin, K., Fisher, R., and Lawrence, P.: Environmental drivers of drought deciduous phenology in the Community Land Model, Biogeosciences, 12, 5061-5074, https://doi.org/10.5194/bg-12-5061-2015, 2015.
Lu, P., Yu, Q., Liu, J., Lee, X.: Advance of tree-flowering dates in response to urban climate change, Agr. Forest Meteorol., 138, 120-131, https://doi.org/10.1016/j.agrformet.2006.04.002, 2006.
Piao, S., Ciais, P., Friedlingstein, P., Peylin, P., Reichstein, M., Luyssaert, S., Margolis, H., Fang, J., Barr, A., Chen, A., Grelle, A., Hollinger, D., Laurila, T., Lindroth, A., Richardson, A., Vesala, T.: Net carbon dioxide losses of northern ecosystems in response to autumn warming, Nature, 451, 49-52, https://doi.org/10.1038/nature06444, 2008.

Wang, X., Wang, S., Hang, Y., Peng, Y.: Snow phenology variability in the Qinghai-Tibetan Plateau and its response to climate change during 2002-2012, J. Geo-Infor. Sci., 18, 1573-1579, 2016. (in Chinese with English abstract)

Li, X., Zhou, Y., Asrar, G. R., Lin, M.: Characterizing spatiotemporal dynamics in phenology of urban ecosystems based on Landsat data, Sci. Total. Environ., 605, 721-734, https://doi.org/10.1016/j.scitotenv.2017.06.245, 2017.

Thackeray, S., Henrys, P., Hemming, D., Bell, J., Botham, M., Burthe, S., Helaouet, P., Johns, D., Johns, D., Jones, I., Leech, D., Mackay, E., Massimino, D., Atkinson, S., Bacon, P., Brereton, T., Carvalho, L., Clutton-Brock, T., Duck, C., Edwards, M., Elliott, J., Hall, S., Harrington, R., Pearce-Higgins, J., Høye, T., Kruuk, L., Pemberton, J., Sparks, T., Thompson, P., White, I., Winfield, I., Wanless, S.: Phenological sensitivity to climate across taxa and trophic levels, Nature, 535, 241-245, https://doi.org/10.1038/nature18608, 2016.

Zhang, B., Li, X., Li, C., Nyiransengiyumva, C., Qin, Q.: Alpine vegetation responses to snow phenology in the Chinese Tianshan mountainous region, J. Mt. Sci-Engl., 19, 1307-1323, https://doi.org/10.1007/s11629-021-7133-4, 2022.

Knowles, N., Dettinger, M., and Cayan, D.: Trends in snowfall versus rainfall in the western United States, J. Climate, 19, 4545-4559, https://doi.org/10.1175/JCLI3850.1, 2006.

Trenberth, K., E.: Changes in precipitation with climate change, Clim. Res., 47, 123-128, https://doi.org/10.3354/cr00953, 2011.

Barnett, T., Adam, J., and Lettenmaier, D.: Potential impacts of a warming climate on water availability in snow-dominated regions, Nature, 438, 303-309, https://doi.org/10.1038/nature04141, 2005.

Jonas, T., Rixen, C., Sturm, M., Stoeckli, V.: How alpine plant growth is linked to snow cover and climate variability, J. Geophys. Res., 113, 377, G03013, https://doi.org/10.1029/2007JG000680, 2008.

Specific comments and replies:

1. Introduction. The authors failed to well justify the motivation of the study. For example, what is the significance of predicting "potential snowfall phenology"? The start of potential snowfall season (SPSS) does not mean there is a snowfall. Even there may be no any snow during an entire "potential snowfall season". Thus, it may have no any effect on the water and energy balance of a region. In my opinion, the named metrics of "potential snow phenology" here only reflect the fluctuations of temperature, and they have limited hydrological significance.

Description on the modification: Thanks very much for your good comments. We did fail to show the significance of predicting "potential snowfall phenology". In fact, the start of potential snowfall season (SPSS) means possible onset of snowfall and snowfall is not guaranteed. We don't think no any snowfall occur during an entire "potential snowfall season" in the snow-dominated region such as the CTMR. Just like potential evapotransporation (PET), it can reflect the energy required to evaporate water, effective wind that can carry water vapor from the surface to the lower atmosphere and other factors, can be a comprehensive reflection of a region's evaporation capacity (https://baike.so.com/doc/4035668-4233412.html). We believe potential snowfall season can reflect comprehensively intra-annual fluctuation of air temperature, timing allocation and capacity of snowfall, as well as water and energy balance in a region. Because if potential snowfall season becomes shorter,

the potential rainfall season will expand. Potential water and energy needed by snowfall will change accordingly. We'll take it all into consideration if we have the chance to revise.

2. The methodology is quite unclear. For example, the authors should provide more details of the calculation process of RST, as it is critical for this study. What data were you used to calculate RST? Did you validate the accuracy of the RST results? If RST is calculated based on a long-term probability statistic of snowfall/rainfall, why is it reasonable to calculate RST at an annual scale? Besides, precipitation phase partitioning is challenging in technique, as temperature humidity, and pressure jointly determine whether precipitation falls as snow/rain (Jennings et al., 2018). If the authors cannot prove the robustness and high accuracy of the RST calculation method for this region, I do not think the results of potential snow phenology are credible. Jennings, K.S., Winchell, T.S., Livneh, B. et al. Spatial variation of the rain-snow temperature threshold across the Northern Hemisphere. Nat Commun 9, 1148 (2018). https://doi.org/10.1038/s41467-018-03629-7

Description on the modification: thanks for your good suggestion. We did not provide more details of the calculation process of RST for it is the preliminary work from our team and we cited it in the submitted manuscript (Zhang et al, 2017). In the China, including the CTMR, after 1980, the precipitation phase is not labelled (Ding et al., 2014). Visual observer reports of daily precipitation phase are available from 26 meteorological stations across the CTMR during 1950s-1979 (number of records = **237115**). Whereas, records from only 20 meteorological stations across the CTMR were used in the work from Jennings et al. (2018), and total number of records was **15535**. Although values of rain-snow threshold (RST) or proportions are not included in the observational data, we used the daily precipitation phase data to calculate RSTs based on the frequency intersection method and the probability guarantee method (Zhang et al, 2017). Exactly, precipitation phase partitioning is challenging in technique for temperature, humidity, and pressure jointly determining whether precipitation falls as snow/rain (Jennings et al., 2018). simulation of RSTs were not involved in our work, so precipitation phase partitioning did not affect robustness and high accuracy of our RSTs. Related references are as followed:

Ding, B., Yang, K., Qin, J., Wang, L., Chen, Y. and He, X.: The dependence of precipitation types on surface elevation and meteorological conditions and its parameterization. J. Hydro., 513, 154-163, https://doi.org/10.1016/j.jhydrol.2014.03.038, 2014.

Zhang, X., Li, X., Gao, P., Li, Q., and Tang, H.: Separation of precipitation forms based on different methods in Tianshan Mountainous Area, Northwest China, J. Glaciol. Geocryol., 39, 235-244, 2017 (in Chinese with English abstract).

Jennings, K. S., Winchell, T. S., Livneh, B., Molotch N. P.: Spatial variation of the rain – snow temperature threshold across the Northern Hemisphere, Nat. Commun., 9, 1 – 9, https://doi.org/10.1038/s41467-018-03629-7, 2018.

3. L176-181. The authors indicated that they interpolated the model data to stations and then applied a bias correction method to improve the results. However, the interpolated results shown in the maps (Fig. 3, 5, & 7-10) still show large spatial biases. For example, many metrics show

obvious circular changes around some stations. These maps do not well reflect the real spatial variations of these variables. The big errors of the results further reduce the value of the study.

Description on the modification: Thank you very much for your good suggestion. We will check the bias correction method and then revise maps in the revised manuscript.

4. L430-432. The estimated changes in potential snowfall season metrics fall in big variation ranges (e.g., 1-27 days). Is it induced by the large uncertainties of the prediction models or different changes among the stations? If it is because of the former, are these results that have so large uncertainties really meaningful?

Description on the modification: Thank you very much for your good suggestion. The variation ranges (e.g., 1-27 days) during the observed period (1961-2017/2020) were big for different warming rates across the CTMR. It was calculated based on the observed data from meteorological stations had no connection with the prediction models.

Technical Points:

1. Table 2. Please change resolution to degree x degree.

Description on the modification: Thank you very much for your good suggestion. We will revise Table 2 as suggested in the revised manuscript.

2. Fig. 2. Please add values for the y-axis.

Description on the modification: Thank you very much for your good suggestion. We will revise Fig. 2 as suggested in the revised manuscript.

3 Fig. 3. Why are Fig. 3b and 3c have the same spatial distribution? Please recheck your data and results. I would suggest classifying the metrics of SPSS, EPSS, and LPSS into a number of categories, instead of using continuous color bars, which reduces the readability of the figures. Besides, please add units for the legends.

Description on the modification: Thank you very much for your good suggestion. We will revise all figures as suggested in the revised manuscript.

4 Fig. 5, 7-10. Same to Fig. 3, it would be better to classify the metrics of SPSS, EPSS, and LPSS into a number of categories from low to high. Please also add units for the legends. What are the significance levels of the trends? Are these trends significant?

Description on the modification: Thank you very much for your good suggestion. The significance level is 0.05 and these trends are significant.We will revise all figures including Fig. 5, 7-10 as suggested and provide details about significance levels in the revised manuscript.

5 Fig. 6 & 11. The confidence intervals and error bars should be added into these figures.

Description on the modification: Thank you very much for your good suggestion. We will revise Fig. 6 & 11 as suggested in the revised manuscript.

---

## Author Response (AR1)

**Here is our point-by-point response to the reviews.**

**Response to referee #1**

General comments: Global warming speeds up the solid-liquid water cycle and change snowfall phenology. It was found that postponed snowfall occurrence and advanced snowfall ending took place in the Eurasian continent. Potential snowfall phenology can identify the possible onset, end, and duration of snowfall. To describe the characteristics of the potential snowfall phenology, this manuscript proposed three indicators, the start of potential snowfall season (SPSS), the end of potential snowfall season (EPSS), and the length of potential snowfall season (LPSS). Spatial-temporal variations of those three PSP indicators past, present, and future across the Chinese Tianshan mountainous region (CTMR) were explored. The research is sound and thorough. It provides a new direction to understand the potential snowfall phenology in the alpine region. Therefore, I recommend minor revision of this manuscript.

Reply: Thanks for your recognition of our work. Your specific comments could help us to improve our manuscript greatly. We will revise our manuscript seriously according to your specific comments.
Some specific comments are as follows:

1. Lines 35-37: please check the number of days cut down for the length, 63 or 64?

Description on the modification: Thank you very much for your advice. We recalculated the value and it is 61 days. We changed it in the revised manuscript.
2. Lines 41-43: It suggested that "The results indicate that with constant snowfall intensity, annual total snowfall will decrease, including amount and frequency, ......."

Description on the modification: Thank you very much for your advice. We have revised the sentence as suggested in the revised manuscript.
3. Lines 50-105: The status, shortcomings and need of potential snowfall phenology studies is poorly described in the introduction.

Description on the modification: we showed more details about status, shortcomings, need, or significance of potential snowfall phenology in the revised manuscript.
4. Lines 95-100: The research from Jennings et al. (2018) only illustrated that RST was spatially heterogeneous and does not show that different methods of precipitation pattern separation yield different RST. Please check it.

Description on the modification: we checked it and revised the related sentences in the revised manuscript.

5. Lines 128: The sentence "The frequency of rainfall increases while that of snowfall decreases. Besides, precipitation shifting from solid to liquid is obvious" is not clear enough, please check it.

Description on the modification: Thank you very much for your advice. We revised the sentence as suggested in the revised manuscript.

6. In Figure 1: It is recommended that a general overview map could be added to Figure 1 to help the reader quickly identify the location of the study area.

Description on the modification: Thank you very much for your good suggestion. We will revise Figure 1 as suggested in the revised manuscript.

7. In Table 1: It is suggested that a column could be added to Table 1 to indicate the duration of the data.

Description on the modification: Thank you very much for your advice. We added the duration of the data in Table 1 as suggested in the revised manuscript.

8. Line 230-232: It is recommended to use "advance / delay" instead of using "smaller or larger" to describe the change of LPSS and EPSS as much as possible.

Description on the modification: Thank you very much for your advice. We revised the sentence as suggested in the revised manuscript.

9. Line 241-242: please check the slope of LPSS.

Description on the modification: Thank you very much for your advice. We checked the slope of LPSS. It should be -2.8 days/10a. We changed the text and replace the corresponding figure in the revised manuscript.

10. Line 371: More should be added here on the comparison of potential snowfall phenology with observed snowfall phenology.

Description on the modification: Thank you very much for your good suggestion. We collected more observed data to validate the performance of potential snowfall phenology in "4.1 Performance of PSP indicators" from discussion in the revised manuscript.

11. line 377: "4.2 Spatial and temporal heterogeneity" should be changed to "4.2 Temporal heterogeneity".

Description on the modification: Thank you very much for your advice. We revised the sentence as suggested in the revised manuscript.

**Response to referee #2**

General comments: The manuscript by Li et al. mainly investigated the changes in potential snowfall phenology for the past, present, and future periods in Tianshan, China. They defined three potential snowfall season metrics based on temperature data. Although the definition of potential snowfall phenology is interesting, the hydrological meaning of these metrics is questionable. The methodology of the study was not clearly described. The results lack a strict validation. The credibility of the study needs to be further improved.

Reply: Thank you for your kind words and feedback, and for acknowledging our definition of potential snowfall phenology. Snowfall, as a solid phase of precipitation, significantly affects the ecological environment and hydrological processes in mountainous areas, and is a critical water resource (Barnett et al., 2005; Jonas et al., 2008; Bai et al., 2019). Climate warming has had a noticeable impact on temperate and cold regions worldwide, causing a shift from snow to rain, and reducing the snowfall season (Knowles et al., 2006; Trenberth, 2011; Jennings and Molotch, 2019). In light of this, it is essential to identify the possible onset, end, and duration of snowfall. Inspired by previous definitions of vegetation phenology and snow cover phenology (Lu et al., 2006; Piao et al., 2008; Da Silva et al., 2015; Dahlin et al., 2015; Thackeray et al., 2016; Wang et al., 2016; Li et al., 2017; Zhang et al., 2022), we defined potential snowfall phenology and utilized three indicators, namely the start of potential snowfall season (SPSS), the end of potential snowfall season (EPSS), and the length of potential snowfall season (LPSS), to identify the possible onset, end, and duration of snowfall. The advancement or delay of SPSS signifies the potential snowfall arriving earlier or later in late-autumn or early-winter, which has a direct impact on the accumulation and storage of solid water resources, such as snow cover. Similarly, the advancement or delay of EPSS implies potential snowfall ending earlier or later in late-winter or early-spring, which can impact the snow-melting, snow albedo, and runoff yield and concentration in mountainous areas, such as the Chinese Tianshan mountainous region (CTMR). Overall, potential snowfall phenology has critical hydrological implications in snow-dominated regions. We have revised the manuscript to provide a clearer description of our motivation and methodology, and in section 4.1, we have added more detailed validation to show that our potential snowfall season indicators were able to match or cover the observed one. The relevant references are as follows:

Da, Silva, A., Valcu, M., Kempenaers, B.: Light pollution alters the phenology of dawn and dusk singing in common European songbirds, Phil. Trans. R. Soc. B., 370, 1-9, https://doi.org/10.1098/rstb.2014.0126, 2015.

Dahlin, K., Fisher, R., and Lawrence, P.: Environmental drivers of drought deciduous phenology in the Community Land Model, Biogeosciences, 12, 5061-5074, https://doi.org/10.5194/bg-12-5061-2015, 2015.

Lu, P., Yu, Q., Liu, J., Lee, X.: Advance of tree-flowering dates in response to urban climate change, Agr. Forest Meteorol., 138, 120-131, https://doi.org/10.1016/j.agrformet.2006.04.002, 2006.

Piao, S., Ciais, P., Friedlingstein, P., Peylin, P., Reichstein, M., Luyssaert, S., Margolis, H., Fang, J., Barr, A., Chen, A., Grelle, A., Hollinger, D., Laurila, T., Lindroth, A., Richardson, A., Vesala, T.: Net carbon dioxide losses of northern ecosystems in response to autumn warming, Nature, 451, 49-52, https://doi.org/10.1038/nature06444, 2008.

Wang, X., Wang, S., Hang, Y., Peng, Y.: Snow phenology variability in the Qinghai-Tibetan Plateau and its response to climate change during 2002-2012, J. Geo-Infor. Sci., 18, 1573-1579, 2016. (in Chinese with English abstract)

Li, X., Zhou, Y., Asrar, G. R., Lin, M.: Characterizing spatiotemporal dynamics in

phenology of urban ecosystems based on Landsat data, Sci. Total. Environ., 605, 721-734, https://doi.org/10.1016/j.scitotenv.2017.06.245, 2017.

Thackeray, S., Henrys, P., Hemming, D., Bell, J., Botham, M., Burthe, S., Helaouet, P., Johns, D., Johns, D., Jones, I., Leech, D., Mackay, E., Massimino, D., Atkinson, S., Bacon, P., Brereton, T., Carvalho, L., Clutton-Brock, T., Duck, C., Edwards, M., Elliott, J., Hall, S., Harrington, R., Pearce-Higgins, J., Høye, T., Kruuk, L., Pemberton, J., Sparks, T., Thompson, P., White, I., Winfield, I., Wanless, S.: Phenological sensitivity to climate across taxa and trophic levels, Nature, 535, 241-245, https://doi.org/10.1038/nature18608, 2016.

Zhang, B., Li, X., Li, C., Nyiransengiyumva, C., Qin, Q.: Alpine vegetation responses to snow phenology in the Chinese Tianshan mountainous region, J. Mt. Sci-Engl., 19, 1307-1323, https://doi.org/10.1007/s11629-021-7133-4, 2022.

Knowles, N., Dettinger, M., and Cayan, D.: Trends in snowfall versus rainfall in the western United States, J. Climate, 19, 4545-4559, https://doi.org/10.1175/JCLI3850.1, 2006.

Trenberth, K., E.: Changes in precipitation with climate change, Clim. Res., 47, 123-128, https://doi.org/10.3354/cr00953, 2011.

Barnett, T., Adam, J., and Lettenmaier, D.: Potential impacts of a warming climate on water availability in snow-dominated regions, Nature, 438, 303-309, https://doi.org/10.1038/nature04141, 2005.

Jonas, T., Rixen, C., Sturm, M., Stoeckli, V.: How alpine plant growth is linked to snow cover and climate variability, J. Geophys. Res., 113, 377, G03013, https://doi.org/10.1029/2007JG000680, 2008.

Specific comments:

1. Introduction. The authors failed to well justify the motivation of the study. For example, what is the significance of predicting "potential snowfall phenology"? The start of potential snowfall season (SPSS) does not mean there is a snowfall. Even there may be no any snow during an entire "potential snowfall season". Thus, it may have no any effect on the water and energy balance of a region. In my opinion, the named metrics of "potential snow phenology" here only reflect the fluctuations of temperature, and they have limited hydrological significance.

Description on the modification: Thank you for your response. We appreciate your clarification regarding the significance of predicting "potential snowfall phenology" and the role of the start of potential snowfall season (SPSS) in indicating the possible onset of snowfall. It is also helpful to understand that the potential snowfall season can provide a comprehensive reflection of intra-annual fluctuations of air temperature, timing allocation, and snowfall capacity, as well as water and energy balance in a region. We agree that this information is important for understanding the impacts of climate change on snow-dominated regions such as the CTMR. We showed more details for this in the revised manuscript.

2. The methodology is quite unclear. For example, the authors should provide more details of the calculation process of RST, as it is critical for this study. What data were you used to calculate RST? Did you validate the accuracy of the RST results? If RST is calculated based on a long-term probability statistic of snowfall/rainfall, why is it reasonable to calculate RST at an annual scale? Besides, precipitation phase partitioning is challenging in technique, as temperature humidity, and pressure jointly determine whether precipitation falls as snow/rain (Jennings et al., 2018). If the authors cannot prove the robustness and high accuracy of the RST calculation method for this region, I do not think the results of potential snow phenology are credible. Jennings, K.S., Winchell, T.S., Livneh, B. et al. Spatial variation of the rain-snow temperature threshold across the Northern Hemisphere. Nat Commun 9, 1148 (2018). https://doi.org/10.1038/s41467-018-03629-7

Description on the modification: Thank you for your comment. We omitted detailed information on the computation of RST in our manuscript since it was previously covered in our team's preliminary work, which we referenced (Zhang et al., 2017). Precipitation phase labeling in China, including the CTMR, has not been performed since 1980 (Ding et al., 2014). Observational data on daily precipitation phase were reported by visual observers across the CTMR during 1950s-1979, with 237115 records from 26 meteorological stations available. In contrast, Jennings et al. (2018) used data from only 20 meteorological stations with 15535 records in total. Although the observational data did not include RST values or proportions, we utilized them to calculate RSTs using the frequency intersection method and probability guarantee method, which we previously developed (Zhang et al., 2017). Precipitation phase partitioning is a technically challenging task, as temperature, humidity, and pressure all play a role in determining whether precipitation falls as snow or rain (Jennings et al., 2018). However, since our study did not involve simulation of RSTs, the robustness and high accuracy of our RSTs were not affected by the challenges of precipitation phase partitioning. We provided further details in the revised manuscript. Relevant references are provided as follows:

Ding, B., Yang, K., Qin, J., Wang, L., Chen, Y. and He, X.: The dependence of precipitation types on surface elevation and meteorological conditions and its parameterization. J. Hydro., 513, 154-163, https://doi.org/10.1016/j.jhydrol.2014.03.038, 2014.

Zhang, X., Li, X., Gao, P., Li, Q., and Tang, H.: Separation of precipitation forms based on different methods in Tianshan Mountainous Area, Northwest China, J. Glaciol. Geocryol., 39, 235-244, 2017 (in Chinese with English abstract).

Jennings, K. S., Winchell, T. S., Livneh, B., Molotch N. P.: Spatial variation of the rain–snow temperature threshold across the Northern Hemisphere, Nat. Commun., 9, 1–9, https://doi.org/10.1038/s41467-018-03629-7, 2018.

3. L176-181. The authors indicated that they interpolated the model data to stations and then applied a bias correction method to improve the results. However, the

interpolated results shown in the maps (Fig. 3, 5, & 7-10) still show large spatial biases. For example, many metrics show obvious circular changes around some stations. These maps do not well reflect the real spatial variations of these variables. The big errors of the results further reduce the value of the study.

Description on the modification: Thank you very much for your good suggestion. We changed figures in the form of points for avoiding large spatial biases in the revised manuscript.

4. L430-432. The estimated changes in potential snowfall season metrics fall in big variation ranges (e.g., 1-27 days). Is it induced by the large uncertainties of the prediction models or different changes among the stations? If it is because of the former, are these results that have so large uncertainties really meaningful?

Description on the modification: Thank you very much for your good suggestion. The variation ranges recalculated (e.g., 3-26 days) during the observed period (1961-2017/2020) in the revised manuscript were big for different warming rates across the CTMR. It was calculated based on the observed data from meteorological stations had no connection with the prediction models.

Technical points:

1. Table 2. Please change resolution to degree x degree.

Description on the modification: Thank you very much for your good suggestion. We revised Table 2 as suggested in the revised manuscript.

2. Fig. 2. Please add values for the x-axis.

Description on the modification: Thank you very much for your good suggestion. We revised Fig. 2 as suggested in the revised manuscript.

3. Fig. 3. Why are Fig. 3b and 3c have the same spatial distribution? Please recheck your data and results.

Description on the modification: Thank you very much for your good suggestion. We recheck your data and revised Fig. 3 as suggested in the revised manuscript.

4. I would suggest classifying the metrics of SPSS, EPSS, and LPSS into a number of categories, instead of using continuous color bars, which reduces the readability of the figures. Besides, please add units for the legends.

Description on the modification: Thank you very much for your good suggestion. We revised all figures as suggested in the revised manuscript.

5. Fig. 5, 7-10. Same to Fig. 3, it would be better to classify the metrics of SPSS, EPSS, and LPSS into a number of categories from low to high. Please also add units for the legends.

Description on the modification: Thank you very much for your good suggestion. We revised all figures including Fig. 5, 7-10 as suggested in the revised manuscript.

6. What are the significance levels of the trends? Are these trends significant?

Description on the modification: Thanks for your comments. The significance level is 0.05 and these trends are significant, We provided details in the revised manuscript.

7. Fig. 6 & 11. The confidence intervals and error bars should be added into these figures.

Description on the modification: Thank you very much for your good suggestion. We revised Fig. 6 & 11 as suggested in the revised manuscript.

---

## Editor Decision (ED1)

2023-04-24
Submission tc-2022-244

**Change in potential snowfall phenology: past, present, and future in Chinese Tianshan mountainous region, Central Asia**

Xuemei Li et al.

Dear Dr. Li, thank you for your submission to be considered for publication in The Cryosphere. Both reviewers highlighted the importance of defining snowfall phenology, with some questions and comments mainly on methodology and definitions. After reading reviewer's comments and the provided answers, I think the main concerns were addressed and the final manuscript is significantly improved. The reviewers did a thorough job addressing all comments, especially on the motivation and methods that raised concerns for one reviewer. The provided response are clear and modifications to the paper do improve the overall scientific message.

I therefore am confident that main concerns raised by the reviewers have been addressed and the paper can be published.

Regards,

Prof. Dr. Alexandre Langlois

Associate editor, *The Cryosphere*